# VISTA: Visual-Semantic Disentanglement and Dynamic Spatial-Temporal Asynchrony for Brain Decoding

## Abstract

Electroencephalogram (EEG) offers a portable, low-cost, and millisecond-scale window into neural dynamics, making it an attractive alternative to functional magnetic resonance imaging (fMRI) for real-world brain visual decoding. Yet fine-grained visual representations and high-level semantic concepts emerge in distinct temporal intervals and spatial topology connections, creating asynchronous patterns that hinder their joint visual-semantic disentanglement. We present VISTA, an EEG-centric neural decoding framework that disentangles visual-semantic modalities along asynchronous spatial-temporal dimensions. Temporally, VISTA divides EEG into non-overlapping time patches and employs an attention mechanism to assign soft weights to each slice, enhancing EEG to capture the heterogeneous temporal distributions of visual and semantic activations. Spatially, it learns modality-specific brain topology connections and derives spatial representation via low-rank decomposition and normalized Laplacian spectral decomposition. The resulting visual and semantic embeddings are each aligned with CLIP's image and text spaces to leverage rich pretrained knowledge. On the large-scale and widely used EEG-visual dataset THINGS-EEG, VISTA outperforms prior EEG methods in zero-shot object recognition. Moreover, on the magnetoencephalogram (MEG) dataset THINGS-MEG, it demonstrates cross-modal generality beyond EEG, achieving comparable gains. Our results underscore the value of asynchronous, disentangled feature extraction and cross-modal alignment for robust neural decoding. Code and pretrained models will be available.

## 1 Introduction

Human cognition is highly dependent on the ability to perceive and respond to complex visual stimuli, which makes neural visual decoding central problems in neuroscience and machine learning (Kamitani & Tong, 2005). Electroencephalogram (EEG), as a fundamental modality in brain-computer interfaces (BCIs), has attracted increasing attention (Du et al., 2021; Minguillon et al., 2017; Samal & Hashmi, 2024; Liu et al., 2025b). Although functional magnetic resonance imaging (fMRI) has dominated visual decoding research due to its superior spatial resolution and well-established methodologies (Zafar et al., 2015; Horikawa & Kamitani, 2017; Allen et al., 2022; Sun et al., 2024), EEG-centric visual decoding (Jiao et al., 2019; Song et al., 2024; Liu et al., 2025a) is gaining recognition for its advantages, including low cost, portability, and high temporal resolution, making it particularly valuable for real-world applications.

EEG signals capture brain activity by recording electrical oscillations generated by synchronized neural firing, providing insights into the dynamic neural processes underlying visual perception (Krigolson et al., 2015). For example, primary visual areas in the occipital cortex process low-level visual features such as color and texture, while higher-order semantic regions in the temporal cortex interpret object semantics (Grill-Spector & Malach, 2004). Recent studies have demonstrated the potential of EEG for visual decoding, including image reconstruction (Li et al., 2024) and object recognition (Du et al., 2021; Ahmed et al., 2021; Luo et al., 2023; Leong et al., 2023). However, the above approaches are limited to a restricted range of visual categories and struggle with generalization, even for known object recognition tasks. Emerging research (Du et al., 2023) on large-scale EEG dataset highlights the feasibility of zero-shot visual decoding in open-set scenarios.

Figure 1: (A) The diagram of motivation and VISTA framework. (B) Visual reconstructions (Jeep, Pear, Dalmation). Red: ground truth; green: reconstruction from subjects 1-5.

Given the difficulty of understanding complex visual stimuli through EEG, the latest approaches leverage pre-trained vision-language models (VLMs) to generate image embeddings, enabling modality alignment between EEG and images (Song et al., 2024). This approach facilitates the extraction of visual representation from EEG through the strong capabilities of large models. However, due to the complex components (e.g., color, texture, and semantic understanding) of EEG signals, there exists a significant modality gap between image embeddings and EEG representation, as seen in Figure 1.A. This means that the flat EEG-image alignment is insufficient. Further, the visual and semantic components exhibit certain *spatial-temporal asynchrony* (Johannes et al., 1995) *(STA)*. Along the temporal dimension, low-level visual features are processed within the first 100 ms by the occipital cortex, while high-level semantic information emerges between 200-300 ms in temporal and prefrontal regions (Thorpe et al., 1996; VanRullen & Thorpe, 2001). Along the spatial dimension, visual representations are dominated by occipital areas, whereas semantic processing involves temporal and frontal lobes (Barceló et al., 2000). This spatial-temporal differentiation provides a theoretical foundation for feature disentanglement research. In this study, we employ spatial-temporal modeling to capture the temporal-spatial asynchronous representation of visual and semantic components, enabling more precise disentanglement and heterogeneous modality alignment.

We propose VISTA, an EEG-centric framework that explicitly disentangles visual and semantic components to address *STA*. *Temporal asynchrony* is handled by dividing EEG signals into non-overlapping time patches and applying an attention module to adaptively learn the temporal activation windows corresponding to visual and semantic modalities. *Spatial asynchrony* is addressed by learning separate visual and semantic brain networks, then extracting spatial encodings via low-rank decomposition and normalized Laplacian spectral decomposition. Each embedding is aligned with CLIP's image or text space for robust cross-modal transfer. Extensive experiments on THINGS-EEG and THINGS-MEG demonstrate state-of-the-art performance of zero-shot object recognition, underscoring the value of asynchronous, disentangled feature extraction for neural decoding.

Our main contributions can be summarized as follows:

- We propose a visual-semantic disentanglement and alignment framework that independently extracts low-level visual and high-level semantic representations, effectively mitigating the generalization issues caused by modality heterogeneity.

- Building on visual-semantic disentanglement, we incorporate spatial-temporal modeling to capture the dynamic asynchrony between visual and semantic modalities in both temporal and spatial dimensions, enhancing the disentanglement performance.

- We achieve state-of-the-art zero-shot decoding performance on large and diverse EEG and MEG datasets. Extensive experiments validate the effectiveness of VISTA.

## 2 RELATED WORK

Decoding neural responses to visual stimuli is a significant focus in machine learning and neuroscience application. Early fMRI-based methods leveraged fine-grained spatial maps for object

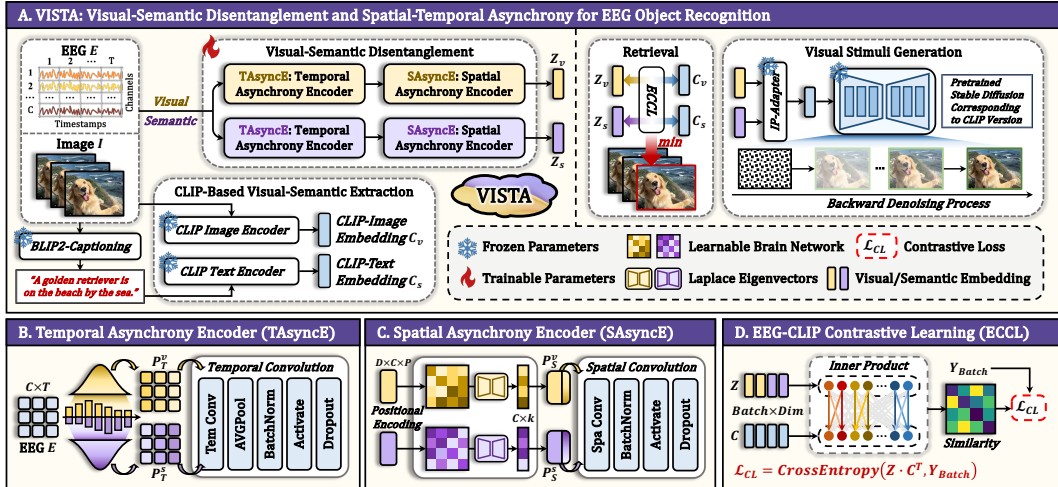

Figure 2: The diagram of VISTA: visual-semantic disentanglement and spatial-temporal asynchrony modeling for neural brain visual decoding.

recognition (Horikawa & Kamitani, 2017), but their low temporal resolution limits real-time applications. EEG, by contrast, captures millisecond-scale dynamics and has shown promise in classifying categories such as faces, animals, and tools (Akamatsu et al., 2020; Leong et al., 2023; Kalafatovich et al., 2020), even extending to unseen classes via large-scale datasets (Luo et al., 2023). Magnetoencephalography (MEG), like EEG, captures neural activity with millisecond precision but offers higher spatial resolution via magnetic field recordings (Zubarev et al., 2019; Boyko et al., 2024), making it a useful complement for studying brain dynamics. Nevertheless, most EEG/MEG approaches remain confined to closed-set scenarios and emphasize visual feature extraction, neglecting higher-level semantics. For example, an EEG dataset (Palazzo et al., 2020) was constructed based on 40 object concepts ImageNet, and further achieved object classification. These methods (Ahmed et al., 2021; Spampinato et al., 2017) identify specific categories (e.g., cat) rather than broad ranges (e.g., animal), limiting generalization. Recent work has introduced zero-shot frameworks using large EEG-visual dataset. THINGS-EEG (Gifford et al., 2022) and THINGS-MEG (Hebart et al., 2023) are released with 1,854 concepts THINGS dataset (Hebart et al., 2019). NICE (Song et al., 2024) and ATM (Li et al., 2024) extended these two datasets to achieve zero-shot recognition of unseen object categories. More recently, an uncertainty-aware approach (Wu et al., 2025) highlighted that intrinsic discrepancies between brain signals and visual stimuli can hinder robust alignment, emphasizing the need for models that explicitly address these mismatches. These methods improve open-set performance but still rely primarily on fine-grained visual embeddings, without explicitly disentangling semantic information. Our proposed VISTA aims to explore the temporal-spatial asynchrony of visual-semantic representations in brain decoding.

## 3 METHOD

Our proposed VISTA disentangles visual and semantic representations while modeling spatial-temporal asynchrony. The overall architecture is illustrated in Figure 2.

### 3.1 PROBLEM DEFINITION

Given a set of EEG-image pairs $X = \{E, I, Y\}$, where $E \in \mathbb{R}^{C \times T}$ denotes the EEG data with $C$ electrode channels and $T$ time samples, $I$ is the corresponding image, and $Y$ represents the ground-true textual caption of the image $I$, our goal is to perform zero-shot object recognition.

During training, the EEG encoder is trained to extract visual and semantic EEG representations that align with image and text embeddings extracted from Contrastive Language-Image Pretraining (CLIP) (Radford et al., 2021). Using EEG-CLIP contrastive learning (ECCL) module, the extracted EEG representations are aligned with the CLIP representation from image and text embeddings,

Figure 3: Overview of VISTA EEG encoder: temporal-spatial asynchrony and visual-semantic disentanglement.

ensuring the encoder to capture both fine-grained visual details and high-level semantic understanding. At test time, recognition is achieved by computing the similarity between the encoded EEG features and CLIP embeddings from unseen object concepts.

### 3.2 FRAMEWORK OVERVIEW

VISTA consists of three core components: EEG encoder, image and text encoder, and contrastive learning module. The EEG encoder performs visual–semantic disentanglement, while the CLIP-based image and text encoders provide large-scale visual and semantic embeddings. Contrastive learning module is used to enforce EEG-CLIP semantic consistency. Specifically, the visual embedding is extracted from the image branch of CLIP, and the semantic embedding is obtained by encoding BLIP-generated captions through the CLIP text encoder. During inference, the visual embedding is used for image retrieval and reconstruction tasks, whereas the semantic embedding is used for semantic retrieval.

### 3.3 EEG ENCODER

The details of the EEG encoder is shown in Figure 3.

#### 3.3.1 TEMPORAL ASYNCHRONY MODELING

To capture temporal asynchrony between visual and semantic EEG responses, we divide the input EEG signal into non-overlapping temporal patches. Unlike previous strategies of directly using the raw EEG data, we introduce a soft gate mechanism to enhance modality-specific patches without discarding information.

EEG data are first divided into $N$ non-overlapping temporal patches $P = \{P_1, P_2, ..., P_N\}$, where $P_i \in \mathbb{R}^{C \times T_p}$ represents localized EEG activity in a fixed temporal window and $T_p = \frac{T}{N}$. For each temporal patch $P_i$, two modality-specific attention modules output continuous importance scores $m_v$ and $m_s$ for visual and semantic relevance, respectively. The process for computing these scores is facilitated by an attention module, which maps input EEG features to patch's importance. Each temporal patch is flattened into a vector $x_i \in \mathbb{R}^F$, where $F = \frac{C \times T}{N}$, and the full patch sequence is represented as $X = \{x_1, x_2, ..., x_N\} \in \mathbb{R}^{N \times F}$. The attention modules project $X$ into query and key spaces to compute a pairwise attention matrix $A^{tem} \in \mathbb{R}^{N \times N}$:

$$Q = XW_q, \quad K = XW_k, \quad A^{tem} = QK^\top \tag{1}$$

where $\{W_q, W_k\} \in \mathbb{R}^{F \times F'}$ are learnable weights, and $A^{tem}_{ij}$ quantifies the relative importance of patch $P_i$ with respect to $P_j$. The attention matrix is then reduced to scalar importance scores $m_i$ for each patch.

Rather than selecting or discarding time series patches, we apply a soft enhancement mechanism to emphasize informative regions. Specifically, each patch is modulated as:

$$P'_i = P_i \cdot (1 + \text{Softplus}(\epsilon) \cdot m_i) \tag{2}$$

where $\epsilon$ is a learnable enhancement factor. This formulation increases the contribution of informative patches while preserving temporal continuity and fine-grained dynamics.

After enhancement, all patches are reassembled into the original temporal sequence $P' \in \mathbb{R}^{C \times T}$. To extract high-level representations, we apply a temporal convolutional module consisting of $D$ convolutional filters of size $1 \times K_T$, followed by average pooling, batch normalization, and a nonlinear activation. This module captures multi-scale temporal dependencies while maintaining spatial structure across EEG channels. The resulting outputs from visual and semantic branches are denoted as $P_T^v, P_T^s \in \mathbb{R}^{D \times C \times T'}$ where $T'$ is the temporal dimensionality after temporal convolution, and are passed into the spatial modeling module for further disentanglement.

### 3.3.2 SPATIAL ASYNCHRONY MODELING

To capture spatial characteristics of EEG responses, we construct learnable visual and semantic brain networks and derive spectral positional encodings to enhance spatial representations.

Specifically, we construct two learnable adjacency matrices $A_v, A_s \in \mathbb{R}^{C \times C}$ that represent visual and semantic brain graphs, respectively. These matrices encode directed, weighted functional connectivity between EEG channels and are optimized throughout visual-semantic disentanglement. These matrices are optimized during training and regularized by applying a non-negativity constraint through ReLU and be further symmetrized as:

$$\bar{A} = \text{ReLU}(A), \quad A_{\text{sym}} = \frac{1}{2}\left(\bar{A} + \bar{A}^\top\right) \tag{3}$$

This ensures the spatial structure captures reciprocal interactions between brain regions when desired.

To incorporate structural information into EEG representations, we derive positional encodings based on the spectrum of the normalized graph Laplacian. Given an adjacency matrix $A$, we compute the normalized graph Laplacian as:

$$L = I - D^{-1/2} A D^{-1/2} \tag{4}$$

where $D$ is the degree matrix and $I$ is the identity matrix. We then perform eigendecomposition of $L$ and retain the first $k$ non-trivial eigenvectors (excluding the zero-eigenvalue vector) to form the positional embedding matrix $P_{\text{pos}}$:

$$P_{\text{pos}} = [\mathbf{e}_1, \mathbf{e}_2, ..., \mathbf{e}_k] \in \mathbb{R}^{C \times k} \tag{5}$$

This spectral basis encodes the intrinsic structure of each brain network in a low-rank form. For each modality, the learned positional encodings are broadcasted and concatenated to the temporal EEG features along the time dimension. Specifically, for an input feature tensor $P_T \in \mathbb{R}^{N \times C \times T'}$, which denotes either the visual or semantic temporal feature representation (i.e., $P_T^v$ or $P_T^s$), we generate a spatially aware representation $P_S \in \mathbb{R}^{N \times C \times (T'+k)}$ by:

$$P_S = \text{Concat}(P_T, P_{\text{pos}}) \tag{6}$$

where $P_{\text{pos}}$ is broadcast across batch and segment dimensions to match input size. The concatenated feature $P_S$ denotes either the visual or semantic temporal feature representation (i.e., $P_S^v$ or $P_S^s$), which then passed through the spatial convolution module. The spatial convolution module composed of $D$ depthwise convolutional filters of size $K_C \times 1$, batch normalization, ELU activation, and dropout. This module aggregates channel-level dependencies while preserving the position-aware information encoded from the graph spectrum. The final outputs from visual and semantic branches, denoted as $Z_V$ and $Z_S$, serve as spatially disentangled EEG features and are passed into downstream projection heads for further alignment.

### 3.4 IMAGE & TEXT ENCODER

For the visual and semantic representations obtained from the EEG encoder, we leverage the CLIP to extract high-quality embeddings from original image and corresponding object text. Specifically, CLIP provides tightly aligned image and text representations, which are crucial for enabling effective contrastive learning and multi-modal alignment.

Given an input image $I$ and an associated object text description $T$, CLIP employs its image encoder $f_{\text{image}}$ and text encoder $f_{\text{text}}$ to generate the corresponding image embedding $C_I = f_{\text{image}}(I)$ and text embedding $C_T = f_{\text{text}}(Y)$, respectively. The advantage of using CLIP lies in its ability to produce semantically rich and aligned embeddings. Unlike other models that extract image and text representations independently, CLIP ensures a tight correspondence between visual and textual representation in a shared latent space.

## 3.5 EEG-CLIP CONTRASTIVE LEARNING

VISTA employs contrastive learning to facilitate the extraction of high-quality representations. Given the EEG-based visual representation $Z_V$ and semantic representation $Z_S$, and the CLIP-generated image embedding $C_I$ and text embedding $C_T$, we define representation pairs $(Z_V, C_I)$ and $(Z_S, C_T)$ for alignment. Inner product of each EEG-CLIP sample pair is computed as the similarity $\text{Sim}(Z, C) = ZC^\top$. Then, InfoNCE loss is employed to minimize the distance between positive pairs while maximizing the distance between negative pairs as follows:

$$\mathcal{L}_{\text{CL}} = \text{CrossEntropy}(\text{Sim}(Z, C), lable_{batch}) \tag{7}$$

where $lable_{batch}$ is the serial number of each sample in the batch. By optimizing the contrastive loss, the EEG embeddings are effectively aligned with their corresponding CLIP embeddings, respectively. The overall loss is formulated as:

$$\mathcal{L}_{\text{VISTA}} = \mathcal{L}_{\text{CL}}^{\text{vis}} + \gamma \mathcal{L}_{\text{CL}}^{\text{sem}} \tag{8}$$

where $\gamma$ are hyperparameters to adjust the weights of $\mathcal{L}_{\text{CL}}^{\text{sem}}$ semantic loss with visual loss component. This loss ensures that disentangled representations effectively align with CLIP embeddings.

# 4 EXPERIMENT

## 4.1 DATASET

We conducted experiments on two large-scale datasets: **THINGS-EEG** (Gifford et al., 2022) and **THINGS-MEG** (Hebart et al., 2023) for comparison.

**THINGS-EEG** is one of the most widely adopted EEG decoding datasets, providing 63-channel EEG recordings from ten participants while viewing images from the **THINGS** (Hebart et al., 2020) database. The training split covers 1,654 object categories, with an additional 200 unseen categories reserved for testing, resulting in 82,160 trials per participant. Signals were band-pass filtered between 0.1–100 Hz, resampled to 100 Hz, segmented into epochs spanning 200 ms before and 800 ms after stimulus onset, and baseline-corrected. To verify generalization, we also tested on **THINGS-MEG**, which contains 271-channel MEG recordings from four participants over 12 sessions. This dataset comprises 1854 object concepts, each paired with 12 unique images for training, and 200 held-out concepts for testing. MEG recordings were epoched from 0–1000 ms, filtered from 0.1–100 Hz, and down-sampled to 200 Hz. More details are provided in Appendix A.2.

## 4.2 EXPERIMENTAL DETAILS

VISTA was implemented in Pytorch and trained on four NVIDIA GTX 3090 GPUs. Hyperparameters include the patch size of 2, size of position embedding $k$ of 4, kernel size $K_T$ for temporal convolution of 25, $K_T$ for spatial convolution of 63, kernel sizes of $1 \times 51$ and $1 \times 5$ for avgpooling, and the $\gamma$ in Eq. (8) is set to 0.8. Training used the ADAM optimizer with a batch size of 1000.

For reproducibility and fairness, object recognition was evaluated under both subject-dependent and subject-independent protocols, consistent with NICE (Song et al., 2024) and ATM (Li et al., 2024), using identical versions of CLIP (CLIP-ViT-H/14[1] (Radford et al., 2021)) and stable diffusion (SDXL-turbo[2] (Podell et al., 2023)). For THINGS-EEG, 740 random training samples were held out as a validation set, and the model checkpoint with the lowest validation loss was used for testing. For THINGS-MEG, we averaged repeated trials for each image, following NICE, to boost signal-to-noise ratio, and compared against state-of-the-art retrieval approaches. Each experiment was repeated five times, and mean performance is reported to account for variance.

## 4.3 OVERALL PERFORMANCE

We selected several recent EEG decoding methods for performance comparison, including DGCNN (Song et al., 2018), EEGNet (Lawhern et al., 2018), EEG-Conformer (Song et al., 2022),

---

[1]https://huggingface.co/laion/CLIP-ViT-H-14-laion2B-s32B-b79K

[2]https://huggingface.co/stabilityai/sdxl-turbo

Table 1: Top-1/5 accuracy (%) on THINGS-EEG for 200-way zero-shot image retrieval.

| Method | Subject 1 | | Subject 2 | | Subject 3 | | Subject 4 | | Subject 5 | | Subject 6 | | Subject 7 | | Subject 8 | | Subject 9 | | Subject 10 | | Ave | |
|---|---|---|---|---|---|---|---|---|---|---|---|---|---|---|---|---|---|---|---|---|---|---|
| | top-1 | top-5 | top-1 | top-5 | top-1 | top-5 | top-1 | top-5 | top-1 | top-5 | top-1 | top-5 | top-1 | top-5 | top-1 | top-5 | top-1 | top-5 | top-1 | top-5 | top-1 | top-5 |
| Subject-dependent (train and test on one subject) | | | | | | | | | | | | | | | | | | | | | | |
| DGCNN [TAFFC'18] | 12.3 | 36.6 | 11.5 | 39.5 | 15.7 | 43.8 | 19.7 | 50.6 | 10.6 | 32.6 | 15.4 | 46.6 | 14.0 | 43.2 | 25.1 | 54.5 | 16.0 | 43.7 | 17.9 | 53.0 | 15.8 | 44.4 |
| EEGNet [JNE'18] | 16.0 | 42.9 | 17.9 | 48.6 | 18.2 | 51.5 | 23.9 | 59.0 | 14.4 | 37.7 | 19.5 | 52.0 | 18.5 | 50.2 | 30.2 | 61.2 | 23.3 | 51.2 | 22.5 | 58.3 | 20.4 | 51.2 |
| EEG-Conformer [TNSRE'22] | 11.4 | 32.4 | 15.2 | 41.9 | 19.8 | 50.9 | 23.0 | 56.6 | 13.6 | 33.4 | 18.1 | 49.0 | 18.5 | 48.2 | 27.1 | 56.9 | 15.2 | 40.0 | 22.6 | 57.8 | 18.4 | 46.7 |
| EEG-ChannelNet [TPAMI'20] | 18.0 | 45.9 | 19.1 | 47.9 | 22.7 | 53.8 | 24.9 | 57.0 | 15.6 | 39.5 | 22.3 | 52.4 | 20.5 | 52.2 | 31.2 | 60.5 | 21.5 | 51.3 | 24.3 | 57.7 | 22.0 | 51.8 |
| Mb2C [ACM MM'24] | 23.6 | 56.3 | 22.6 | 50.5 | 26.3 | 60.1 | 34.8 | 67.0 | 21.3 | 53.0 | 31.0 | 62.3 | 25.0 | 54.8 | 39.0 | 69.3 | 27.5 | 59.3 | 33.1 | 70.8 | 28.4 | 60.3 |
| NICE [ICLR'24] | 21.7 | 51.2 | 23.3 | 55.0 | 29.1 | 60.5 | 32.3 | 69.6 | 18.2 | 45.6 | 29.3 | 62.1 | 24.3 | 59.2 | 41.3 | 72.4 | 24.3 | 59.0 | 28.9 | 62.6 | 27.3 | 59.7 |
| ATM [NeurIPS'24] | 22.9 | 52.7 | 24.5 | 58.5 | 31.7 | 64.6 | 30.3 | 70.1 | 21.4 | 49.0 | 26.5 | 63.4 | 27.5 | 62.0 | 39.6 | 71.9 | **32.9** | 63.6 | 31.8 | 65.6 | 28.9 | 62.1 |
| CognitionCapturer [AAAI'25] | 22.5 | 50.4 | 23.3 | 54.9 | 26.8 | 58.9 | 29.7 | 65.7 | 18.5 | 44.9 | 25.4 | 55.7 | 25.9 | 57.4 | 38.8 | 67.7 | 27.6 | 53.8 | 28.2 | 62.8 | 26.7 | 57.2 |
| EEG-CLIP [NN'25] | 24.2 | 55.1 | 24.9 | 59.1 | 31.0 | 62.9 | 32.1 | 68.4 | 22.3 | 50.2 | 25.2 | 62.5 | 27.9 | 62.3 | 38.9 | 70.5 | 31.6 | 63.1 | 30.2 | 65.2 | 28.8 | 61.9 |
| BrainFLORA [ACM MM'25] | 23.6 | 53.1 | 24.7 | 58.4 | 31.5 | 64.3 | 33.4 | 70.1 | 20.3 | 48.9 | 29.0 | 64.2 | 25.9 | 61.6 | 41.5 | 73.2 | 29.5 | 61.3 | 31.2 | 65.0 | 29.1 | 62.0 |
| SRT [ICCV'25] | 25.9 | 57.7 | 25.2 | 60.8 | 30.4 | 62.8 | 33.5 | 66.8 | 22.8 | 52.0 | 24.1 | 61.6 | 29.8 | 63.9 | 40.1 | 70.2 | 31.5 | **64.4** | 28.7 | 65.7 | 29.2 | 62.6 |
| **VISTA [Ours]** | **26.1** | **57.5** | **28.8** | **61.3** | **34.1** | **66.9** | **38.2** | **72.0** | **24.4** | **52.3** | **31.3** | **67.7** | **28.9** | **66.4** | **46.2** | **74.7** | **31.8** | **63.5** | **34.9** | **68.2** | **32.5** | **65.1** |
| Subject-independent (leave-one-subject-out) | | | | | | | | | | | | | | | | | | | | | | |
| DGCNN [TAFFC'18] | 10.8 | 30.6 | 11.9 | 31.3 | 6.3 | 21.3 | 8.2 | 24.6 | 6.1 | 18.5 | 11.2 | 30.4 | 6.8 | 20.5 | 11.0 | 28.6 | 9.4 | 25.4 | 12.2 | 30.6 | 9.4 | 26.2 |
| EEGNet [JNE'18] | 11.0 | 31.1 | 11.4 | 34.0 | 6.5 | 24.7 | 13.4 | 34.4 | 6.7 | 26.7 | 8.5 | 29.4 | 7.9 | 22.0 | 10.7 | 32.9 | 8.9 | 27.7 | 15.1 | 41.6 | 10.0 | 30.4 |
| EEG-Conformer [TNSRE'22] | 6.3 | 22.3 | 5.7 | 20.4 | 5.8 | 15.7 | 7.8 | 21.8 | 6.7 | 18.4 | 10.4 | 32.9 | 7.0 | 24.1 | 9.1 | 25.2 | 5.0 | 17.2 | 11.7 | 33.2 | 7.5 | 23.1 |
| EEG-ChannelNet [TPAMI'20] | 11.2 | 33.6 | 12.8 | 34.4 | 8.4 | 26.5 | 13.5 | 33.8 | 9.7 | 29.0 | 10.8 | 33.9 | 9.7 | 27.7 | 11.6 | 32.3 | 10.4 | 30.8 | 15.7 | 39.5 | 11.4 | 32.2 |
| Mb2C [ACM MM'24] | 10.5 | 28.1 | 11.3 | 32.8 | 8.8 | 27.6 | 13.6 | 33.5 | 10.6 | 27.5 | 12.1 | 33.1 | 11.5 | 31.8 | 12.0 | 32.1 | 12.1 | 31.3 | 16.1 | 42.1 | 11.9 | 32.0 |
| NICE [ICLR'24] | 13.5 | 39.3 | 16.7 | 42.7 | 12.4 | 35.5 | 17.9 | 41.9 | 14.7 | 38.1 | 15.6 | 44.6 | 13.6 | 39.0 | 13.5 | 37.2 | 17.0 | 42.0 | 22.8 | 50.7 | 15.7 | 41.1 |
| ATM [NeurIPS'24] | 17.1 | 41.8 | 20.2 | 44.2 | 13.2 | 36.7 | 17.0 | 40.7 | 15.1 | 41.0 | 13.5 | 38.3 | 10.1 | 29.0 | 15.2 | 41.9 | 13.5 | 38.4 | 20.0 | 45.4 | 15.5 | 39.6 |
| CognitionCapturer [AAAI'25] | 16.5 | 42.3 | 16.2 | 37.9 | 9.5 | 26.8 | 15.4 | 37.6 | 10.3 | 31.7 | 14.0 | 35.4 | 10.7 | 26.9 | 14.5 | 34.2 | 9.4 | 32.4 | 15.3 | 38.6 | 13.2 | 34.4 |
| EEG-CLIP [NN'25] | 14.6 | 39.6 | 17.4 | 39.6 | 11.0 | 32.8 | 16.1 | 38.9 | 13.4 | 35.3 | 15.3 | 38.1 | 13.1 | 33.2 | 14.3 | 37.6 | 13.8 | 36.2 | 19.5 | 46.7 | 14.9 | 37.8 |
| BrainFLORA [ACM MM'25] | 13.5 | 42.0 | 18.5 | 44.3 | 13.4 | 38.6 | 18.1 | 41.0 | 15.5 | 39.4 | 15.4 | 42.8 | 12.4 | 35.2 | 14.5 | 38.9 | 13.5 | 40.1 | 20.2 | 49.2 | 15.5 | 41.2 |
| SRT [ICCV'25] | 17.3 | 39.8 | 19.4 | 42.4 | 14.5 | 38.8 | 15.8 | 40.2 | 15.9 | 38.1 | 17.1 | 41.4 | 11.7 | 31.1 | 14.9 | 37.0 | 13.4 | 38.3 | 23.5 | 54.6 | 16.4 | 40.2 |
| **VISTA [Ours]** | **17.9** | **45.7** | **22.1** | **49.5** | **17.5** | **44.8** | **20.4** | **48.5** | **16.8** | **41.2** | **20.1** | **49.6** | **15.2** | **42.5** | **17.6** | **40.7** | **18.3** | **42.7** | **25.1** | **55.4** | **19.1** | **46.1** |

Table 2: Top-1/5 accuracy (%) on THINGS-EEG for 200-way zero-shot semantic retrieval.

| Method | Subject 1 | | Subject 2 | | Subject 3 | | Subject 4 | | Subject 5 | | Subject 6 | | Subject 7 | | Subject 8 | | Subject 9 | | Subject 10 | | Ave | |
|---|---|---|---|---|---|---|---|---|---|---|---|---|---|---|---|---|---|---|---|---|---|---|
| | top-1 | top-5 | top-1 | top-5 | top-1 | top-5 | top-1 | top-5 | top-1 | top-5 | top-1 | top-5 | top-1 | top-5 | top-1 | top-5 | top-1 | top-5 | top-1 | top-5 | top-1 | top-5 |
| Subject-dependent (train and test on one subject) | | | | | | | | | | | | | | | | | | | | | | |
| DGCNN [TAFFC'18] | 8.6 | 23.7 | 7.8 | 25.1 | 8.1 | 27.2 | 11.2 | 36.9 | 5.6 | 19.2 | 9.2 | 29.6 | 9.6 | 31.5 | 12.4 | 39.3 | 9.3 | 27.7 | 11.9 | 32.6 | 9.4 | 29.3 |
| EEGNet [JNE'18] | 8.7 | 27.9 | 9.4 | 29.8 | 10.6 | 28.7 | 12.2 | 38.9 | 8.1 | 25.3 | 9.6 | 30.4 | 9.1 | 33.3 | 14.3 | 37.8 | 11.1 | 30.2 | 12.9 | 36.2 | 10.6 | 31.9 |
| EEG-Conformer [TNSRE'22] | 7.4 | 22.3 | 6.7 | 25.4 | 9.6 | 30.3 | 14.0 | 38.7 | 7.3 | 21.9 | 9.1 | 30.2 | 9.1 | 31.7 | 13.8 | 40.7 | 10.2 | 30.3 | 13.1 | 36.7 | 10.1 | 30.8 |
| EEG-ChannelNet [TPAMI'20] | 8.7 | 29.8 | 9.6 | 30.9 | 10.2 | 32.3 | 14.8 | 40.4 | 8.0 | 25.6 | 10.5 | 33.3 | 10.6 | 35.5 | 14.9 | 40.5 | 11.5 | 31.7 | 13.3 | 36.9 | 11.2 | 33.7 |
| Mb2C [ACM MM'24] | 10.0 | 35.5 | 9.6 | 33.8 | 12.2 | 38.9 | 18.5 | 46.2 | 8.9 | 30.8 | 13.2 | 40.9 | 11.4 | 40.5 | 20.1 | 52.3 | 13.2 | 39.3 | 15.1 | 41.8 | 13.2 | 40.0 |
| NICE [ICLR'24] | 9.7 | 34.4 | 9.8 | 34.0 | 13.4 | 39.1 | 17.5 | 48.5 | 9.1 | 30.0 | 14.3 | 40.2 | 12.6 | 40.1 | 19.4 | 52.1 | 13.8 | 38.1 | 14.2 | 40.5 | 13.4 | 39.7 |
| ATM [NeurIPS'24] | 13.3 | 37.3 | 13.4 | 38.8 | 13.7 | 40.6 | 18.5 | 47.2 | 11.1 | 34.1 | 15.2 | 42.1 | 15.4 | 46.7 | 22.1 | 54.1 | 13.8 | 39.8 | 18.5 | 43.6 | 15.5 | 42.4 |
| CognitionCapturer [AAAI'25] | 7.9 | 36.9 | 13.8 | **40.7** | 12.4 | 38.4 | 18.8 | 51.8 | 8.1 | 33.2 | 12.5 | 46.4 | 13.8 | 44.1 | 21.5 | 55.8 | 15.8 | 43.7 | 17.6 | 44.0 | 14.5 | 43.5 |
| EEG-CLIP [NN'25] | 12.7 | 38.6 | 12.3 | 38.2 | 15.9 | 41.7 | 19.5 | 49.8 | 10.5 | 34.1 | 14.8 | 43.1 | 15.0 | 44.6 | 21.7 | 55.7 | 16.4 | 41.0 | 18.1 | 43.5 | 15.7 | 42.8 |
| BrainFLORA [ACM MM'25] | 10.3 | 36.7 | 13.3 | 38.1 | 14.1 | 39.1 | 19.7 | 48.0 | 10.4 | 32.7 | 14.7 | 43.3 | 13.6 | 42.4 | 19.2 | 52.4 | 16.2 | 40.3 | 17.4 | 42.8 | 14.9 | 41.6 |
| SRT [ICCV'25] | 11.8 | 37.6 | **14.2** | 39.0 | 14.7 | 42.0 | 19.8 | 49.2 | 11.5 | 33.4 | 16.8 | 44.2 | 14.7 | 44.9 | 22.5 | 56.4 | 15.7 | 41.5 | 18.5 | 43.2 | 16.2 | 43.1 |
| **VISTA [Ours]** | **14.5** | **42.4** | 13.6 | **40.1** | **16.6** | **44.1** | **21.9** | **52.4** | **13.3** | **34.4** | **18.2** | **45.8** | **16.9** | **48.6** | **27.0** | **58.4** | **18.7** | **46.3** | **19.8** | **46.5** | **18.1** | **45.9** |
| Subject-independent (leave-one-subject-out) | | | | | | | | | | | | | | | | | | | | | | |
| DGCNN [TAFFC'18] | 6.3 | 21.1 | 6.8 | 24.7 | 5.2 | 19.9 | 5.9 | 22.4 | 4.8 | 17.3 | 7.1 | 21.4 | 4.7 | 18.2 | 7.3 | 23.5 | 6.4 | 23.6 | 7.5 | 24.3 | 6.2 | 21.7 |
| EEGNet [JNE'18] | 5.5 | 20.4 | 4.5 | 20.5 | 5.7 | 21.5 | 6.3 | 24.3 | 4.3 | 17.3 | 6.6 | 23.6 | 5.8 | 19.4 | 6.8 | 22.3 | 4.3 | 16.4 | 9.0 | 28.5 | 5.9 | 21.4 |
| EEG-Conformer [TNSRE'22] | 4.4 | 17.1 | 4.1 | 18.3 | 4.8 | 19.8 | 6.7 | 20.6 | 3.2 | 16.2 | 5.4 | 17.8 | 4.4 | 18.6 | 6.5 | 20.9 | 3.2 | 15.9 | 9.3 | 25.9 | 5.2 | 19.1 |
| EEG-ChannelNet [TPAMI'20] | 7.5 | 23.2 | 6.9 | 23.4 | 6.2 | 23.2 | 8.9 | 27.1 | 5.2 | 19.1 | 6.6 | 21.1 | 8.5 | 24.4 | 5.5 | 19.2 | 10.2 | 30.2 | | | 7.4 | 23.6 |
| Mb2C [ACM MM'24] | 9.6 | 31.3 | 11.5 | 29.9 | 9.2 | 32.1 | 10.6 | 30.8 | 8.0 | 26.2 | 11.0 | 28.6 | 8.3 | 21.2 | 10.9 | 30.5 | 9.8 | 27.0 | 13.2 | 34.4 | 10.8 | 29.3 |
| NICE [ICLR'24] | 10.3 | 26.8 | 9.2 | 29.7 | 8.9 | 32.4 | 13.1 | 35.4 | 7.5 | 24.7 | 10.1 | 29.2 | 9.8 | 28.8 | 11.2 | 28.8 | 8.8 | 27.6 | 16.3 | 42.1 | 10.5 | 30.6 |
| ATM [NeurIPS'24] | 9.6 | 32.5 | **12.2** | 30.1 | 9.4 | 32.1 | 10.1 | 29.8 | 8.3 | 26.7 | 11.4 | 28.6 | 8.7 | 24.5 | 11.1 | 30.9 | 10.2 | 27.0 | 12.6 | 32.6 | 10.4 | 29.5 |
| CognitionCapturer [AAAI'25] | 10.1 | 28.2 | 9.6 | 30.7 | 9.9 | 34.5 | 12.5 | 34.1 | 8.3 | 27.5 | 9.3 | 29.6 | 9.4 | 27.8 | 12.5 | 30.6 | 9.0 | 26.2 | 16.8 | 42.7 | 10.7 | 31.2 |
| EEG-CLIP [NN'25] | 10.6 | 33.7 | 12.7 | 33.0 | 10.9 | 33.7 | 11.5 | 33.2 | 9.0 | 28.8 | 12.4 | 30.3 | 10.1 | 30.3 | 12.5 | 32.3 | 10.5 | 29.1 | 13.9 | 39.6 | 11.4 | 32.4 |
| BrainFLORA [ACM MM'25] | 9.8 | 29.4 | 10.1 | 31.5 | 9.2 | 31.3 | 12.2 | 34.5 | 8.7 | 24.7 | 9.3 | 29.1 | 9.4 | 27.5 | 11.4 | 29.9 | 9.5 | 27.8 | 15.0 | 39.2 | 10.5 | 30.5 |
| SRT [ICCV'25] | 10.9 | 30.1 | 11.1 | 31.2 | 10.8 | 33.5 | 12.2 | 33.8 | 9.5 | 27.4 | 11.1 | 30.5 | 9.7 | 29.6 | 12.7 | 31.8 | 10.3 | 28.5 | 14.9 | 41.4 | 11.3 | 31.8 |
| **VISTA [Ours]** | **13.6** | **33.9** | **11.7** | **36.8** | **15.4** | **37.1** | **13.6** | **36.9** | **10.7** | **31.5** | **12.7** | **32.9** | **10.6** | **32.2** | **14.2** | **34.3** | **11.0** | **31.1** | **19.7** | **46.8** | **13.4** | **35.3** |

EEG-ChannelNet (Palazzo et al., 2020), Mb2C (Wei et al., 2024), NICE (Song et al., 2024), ATM (Li et al., 2024), CognitionCapturer (Zhang et al., 2024), EEG-CLIP (Cao et al., 2025), BrainFLORA (Li et al., 2025b), and SRT (Kim et al., 2025). Retaining the same evaluation approach as NICE/ATM, we conducted comparisons, and the results are presented in Table 1, Table 2, and Table 3.

Table 1 reports the top-1/5 accuracies on THINGS-EEG for 200-way zero-shot image retrieval under both subject-dependent and subject-independent settings. Across all ten subjects, VISTA consistently achieves the best performance, reaching an average top-1 accuracy of 32.5% and top-5 accuracy of 65.1%, which represent clear improvements over prior state-of-the-art methods such as ATM (top-1 28.9%, top-5 62.1%) and NICE (top-1 27.3%, top-5 59.7%). The performance gain is especially pronounced for subject-dependent evaluation, where VISTA exceeds ATM by $+3.6\%$ top-1 accuracy on average, and by up to $+6.6\%$ for the most challenging Subject 8. Similar trends are observed in the subject-independent setting, where VISTA improves the average top-1 accuracy from 15.7% (NICE) to 19.1%, indicating that the proposed approach generalizes better across unseen participants.

Semantic retrieval results on THINGS-EEG, summarized in Table 2, exhibit a comparable trend. VISTA again achieves the highest average top-1 accuracy (18.1%) and top-5 accuracy (45.9%) in the subject-dependent setting, outperforming ATM and CognitionCapturer by $+2.6\%$ and $+3.6\%$ top-1 respectively. The performance advantage remains consistent in leave-one-subject-out evalu-

Table 3: Top-1/5 accuracy (%) on THINGS-MEG for 200-way zero-shot visual/semantic retrieval.

| Method | Subject 1 | | Subject 2 | | Subject 3 | | Subject 4 | | Ave | | Subject 1 | | Subject 2 | | Subject 3 | | Subject 4 | | Ave | |
|---|---|---|---|---|---|---|---|---|---|---|---|---|---|---|---|---|---|---|---|---|
| | top-1 | top-5 | top-1 | top-5 | top-1 | top-5 | top-1 | top-5 | top-1 | top-5 | top-1 | top-5 | top-1 | top-5 | top-1 | top-5 | top-1 | top-5 | top-1 | top-5 |
| Subject-dependent (train and test on one subject) | | | | | | | | | | | | | | | | | | | | |
| DGCNN [TAFFC'18] | 6.7 | 22.0 | 14.1 | 36.2 | 11.9 | 35.8 | 7.7 | 22.8 | 10.1 | 29.2 | 5.7 | 17.7 | 8.6 | 29.2 | 10.8 | 30.2 | 6.9 | 20.3 | 8.0 | 24.4 |
| EEGNet [JNE'18] | 9.8 | 29.2 | 17.9 | 48.9 | 14.8 | 41.3 | 9.1 | 28.7 | 12.9 | 37.0 | 6.4 | 21.4 | 12.8 | 36.2 | 13.6 | 33.1 | 5.5 | 20.4 | 9.6 | 27.8 |
| EEG-Conformer [TNSRE'22] | 9.1 | 27.1 | 20.8 | 47.4 | 15.2 | 40.7 | 9.8 | 27.9 | 13.7 | 35.8 | 6.0 | 22.2 | 18.8 | 38.2 | 13.7 | 33.6 | 7.1 | 18.8 | 9.7 | 28.2 |
| EEG-ChannelNet [TPAMI'20] | 10.8 | 33.0 | 22.6 | 52.2 | 18.5 | 45.2 | 10.4 | 32.6 | 15.6 | 40.8 | 7.7 | 24.4 | 13.8 | 39.9 | 14.5 | 35.1 | 7.2 | 22.6 | 10.8 | 30.5 |
| Mb2C [ACM MM'24] | 9.3 | 33.6 | 20.6 | 49.2 | 18.2 | 44.3 | 10.2 | 33.6 | 14.6 | 39.9 | 8.6 | 25.7 | 12.8 | 37.0 | 13.2 | 34.1 | 8.2 | 22.6 | 10.7 | 29.9 |
| NICE [ICLR'24] | 11.5 | 35.6 | 25.7 | 54.4 | 21.0 | 47.8 | 11.2 | 35.2 | 17.4 | 43.3 | 8.8 | 27.3 | 15.0 | 43.2 | 15.5 | 37.3 | 8.7 | 24.9 | 12.0 | 33.2 |
| ATM [NeurIPS'24] | 8.0 | 29.3 | **30.2** | **61.5** | 20.3 | 50.5 | 11.8 | 33.3 | 17.6 | 43.7 | 6.9 | 20.9 | 16.9 | 46.1 | 15.0 | 39.8 | 7.5 | 26.4 | 11.6 | 33.3 |
| CognitionCapturer [AAAI'25] | 10.1 | 30.6 | 26.9 | 55.1 | 20.4 | 49.3 | 11.5 | 34.2 | 17.2 | 42.3 | 7.5 | 23.6 | 15.3 | 43.7 | 14.7 | 37.1 | 8.0 | 25.6 | 11.4 | 32.5 |
| **VISTA** [Ours] | **11.7** | **34.9** | 26.4 | 56.5 | **22.0** | **53.1** | **12.2** | **35.3** | **18.1** | **44.9** | **10.3** | **30.6** | **17.7** | **44.5** | **16.8** | **38.4** | **8.2** | **26.5** | **13.3** | **35.0** |

Table 4: Ablation study: Top-1/5 accuracy (%) on THINGS-EEG for 200-way zero-shot visual/semantic retrieval. (Yellow: visual retrieval, purple: semantic retrieval.)

| Method | Subject 1 | | Subject 2 | | Subject 3 | | Subject 4 | | Subject 5 | | Subject 6 | | Subject 7 | | Subject 8 | | Subject 9 | | Subject 10 | | Ave | |
|---|---|---|---|---|---|---|---|---|---|---|---|---|---|---|---|---|---|---|---|---|---|---|
| | top-1 | top-5 | top-1 | top-5 | top-1 | top-5 | top-1 | top-5 | top-1 | top-5 | top-1 | top-5 | top-1 | top-5 | top-1 | top-5 | top-1 | top-5 | top-1 | top-5 | top-1 | top-5 |
| Subject-dependent (train and test on one subject) | | | | | | | | | | | | | | | | | | | | | | |
| Temporal-Only | 25.8 | 56.6 | 29.2 | 58.3 | 33.0 | 66.2 | 35.5 | 71.9 | 23.4 | 51.8 | 29.6 | 65.8 | 28.2 | 62.7 | 44.5 | 74.3 | 29.7 | 63.6 | 31.4 | 66.3 | 31.0 | 63.8 |
| Spatial-Only | 25.6 | 57.4 | 27.8 | 59.1 | 32.7 | 63.8 | **38.4** | 71.6 | 24.1 | **52.7** | 30.5 | 63.3 | 28.4 | 62.6 | 44.0 | 73.1 | 31.4 | 63.5 | 32.4 | 67.0 | 31.5 | 63.4 |
| Both-Off | 23.3 | 54.5 | 26.5 | 57.8 | 30.4 | 64.5 | 36.1 | 70.6 | 20.9 | 46.6 | 29.8 | 62.7 | 27.2 | 60.9 | 40.0 | 71.5 | 28.2 | 59.8 | 31.3 | 63.7 | 29.4 | 61.3 |
| Visual-Only | **26.7** | 57.3 | 28.1 | 60.2 | 33.5 | 66.7 | 37.7 | 71.3 | **24.8** | 51.6 | 30.2 | 65.4 | 28.7 | 65.7 | 43.5 | 72.8 | 31.5 | **63.8** | 33.5 | 68.1 | 31.8 | 64.3 |
| **VISTA** | 26.1 | **57.5** | **28.8** | **61.3** | **34.1** | **66.9** | 38.2 | **72.0** | 24.4 | 52.3 | **31.3** | **67.7** | **28.9** | **66.4** | **46.2** | **74.7** | **31.8** | 63.5 | **34.9** | **68.2** | **32.5** | **65.1** |
| Temporal-Only | 14.3 | 40.7 | **14.3** | 38.6 | 16.1 | 43.8 | 21.1 | 52.4 | 12.7 | **34.4** | 17.1 | **46.4** | 16.2 | 46.4 | 24.8 | 58.2 | 18.3 | 46.3 | 19.1 | 46.8 | 17.4 | 45.4 |
| Spatial-Only | 13.5 | 39.7 | 13.9 | 38.4 | 15.4 | **44.1** | 22.6 | **53.2** | 11.8 | 33.1 | 16.1 | 45.4 | 16.6 | 45.7 | 24.1 | 58.0 | 17.7 | 44.2 | **20.3** | **48.5** | 17.2 | 45.0 |
| Both-Off | 13.1 | 37.7 | 11.2 | 36.2 | 13.5 | 42.6 | 21.5 | 51.5 | 9.0 | 30.5 | 15.6 | 44.8 | 15.5 | 45.5 | 22.3 | 57.1 | 16.9 | 42.1 | 18.7 | 46.0 | 15.7 | 43.4 |
| Semantic-Only | 14.3 | 41.8 | 12.8 | **40.1** | 16.7 | 43.3 | 21.1 | 50.9 | 12.4 | 32.8 | **16.8** | 43.6 | 16.8 | 47.1 | 25.5 | 56.6 | 16.9 | **46.6** | 19.3 | 47.7 | 17.3 | 45.1 |
| **VISTA** | **14.5** | **42.3** | 13.1 | 39.5 | **16.3** | 43.9 | 21.6 | 52.2 | **13.0** | 34.1 | **17.9** | 45.5 | 16.7 | **48.5** | **26.7** | **58.3** | **18.4** | 46.2 | 19.5 | 46.3 | **17.8** | **45.7** |
| Subject-independent (leave-one-subject-out) | | | | | | | | | | | | | | | | | | | | | | |
| Temporal-Only | 17.1 | 43.5 | 20.1 | 46.9 | **17.7** | 44.4 | 19.2 | 44.7 | 15.5 | 39.6 | 19.1 | 47.3 | 14.8 | 40.6 | 16.3 | 40.5 | 16.1 | **42.9** | 25.6 | **55.7** | 18.2 | 44.6 |
| Spatial-Only | **18.1** | 45.3 | 18.6 | **50.1** | 17.6 | 43.2 | **20.8** | **48.8** | 16.6 | 40.3 | 19.5 | 48.2 | 14.8 | 42.1 | 16.5 | 40.4 | 15.7 | 41.0 | 25.8 | 55.0 | 18.4 | 45.4 |
| Both-Off | 15.8 | 41.5 | 19.8 | 45.6 | 15.1 | 39.7 | 19.7 | 45.5 | 16.5 | 39.4 | 18.5 | 48.4 | 13.5 | 40.0 | 15.3 | **41.3** | 16.7 | 42.3 | **26.4** | 54.6 | 17.7 | 43.8 |
| Visual-Only | 17.3 | 44.2 | 20.5 | 49.2 | 17.5 | 44.1 | 18.9 | 48.0 | 16.2 | 40.7 | 19.6 | 47.8 | 15.1 | 41.1 | 17.2 | 41.0 | 18.5 | 42.3 | 24.4 | 53.2 | 18.5 | 45.2 |
| **VISTA** | 17.9 | **45.7** | **22.1** | 49.5 | 17.5 | **44.8** | 20.4 | 48.5 | **16.8** | 41.2 | **20.1** | 49.6 | 15.2 | 42.5 | 17.6 | 40.7 | **18.3** | 42.7 | 25.1 | 55.4 | **19.1** | 46.1 |
| Temporal-Only | 12.3 | 33.8 | **12.1** | 35.8 | 13.1 | 36.5 | 13.5 | 36.2 | 10.3 | 30.5 | **13.2** | 31.9 | 10.9 | 30.8 | 13.3 | 33.7 | 10.8 | 29.9 | 18.6 | 44.1 | 12.8 | 34.2 |
| Spatial-Only | 12.4 | 32.2 | 11.5 | 36.3 | **15.7** | 36.4 | **14.1** | **39.5** | 9.1 | 28.7 | 11.4 | **33.8** | 10.5 | 30.5 | 13.0 | 34.4 | 10.5 | 28.9 | 18.9 | **48.1** | 12.7 | 34.9 |
| Both-Off | 11.9 | 33.0 | 11.3 | 33.2 | 11.6 | 33.8 | 13.7 | 37.1 | 9.8 | 27.5 | 11.8 | 32.6 | 9.6 | 30.3 | 13.4 | 32.0 | 10.8 | **31.4** | 19.1 | 47.5 | 12.2 | 33.7 |
| Semantic-Only | 12.5 | **34.1** | 11.5 | 34.6 | 14.2 | 36.1 | 13.6 | 36.2 | 10.4 | 29.8 | 12.8 | 33.2 | 10.1 | 31.5 | 14.0 | 32.2 | 10.6 | 29.7 | 17.5 | 46.5 | 12.7 | 34.4 |
| **VISTA** | 13.6 | 33.9 | 11.7 | **36.8** | 15.4 | **37.1** | 13.6 | 36.9 | **10.7** | **31.5** | 12.7 | 32.9 | 10.6 | 32.2 | 14.2 | 34.3 | **11.0** | 31.1 | **19.7** | 46.8 | **13.4** | **35.3** |

ation, suggesting that VISTA not only captures fine-grained visual representations but also learns discriminative semantic embeddings that are transferable across subjects.

We further evaluate on THINGS-MEG to verify the robustness of the proposed approach under a different neuroimaging modality. As shown in Table 3, VISTA yields the highest mean top-1 and top-5 accuracies among all compared methods, demonstrating its effectiveness in leveraging spatiotemporal neural dynamics beyond EEG. Overall, these results confirm that VISTA achieves state-of-the-art performance on both visual and semantic retrieval tasks, generalizes well to unseen subjects, and transfers robustly across neuroimaging modalities.

## 4.4 ABLATION STUDY

We conducted ablation experiments to evaluate the contribution of each module in VISTA, as shown in Table 4. **More details of ablation study and parameter analysis are provided in Appendix A.4.**

### 4.4.1 ABLATION ON VISUAL-SEMANTIC DISENTANGLEMENT

To assess the contribution of visual and semantic pathways, we evaluated the disentanglement module under two simplified configurations: 1) Visual-Only, where only visual representations were used for alignment, and 2) Semantic-Only, where only semantic embeddings participated in training. As shown in Table 4, both configurations yielded inferior performance compared to the full VISTA model in both subject-dependent and subject-independent settings. In the subject-dependent setting, VISTA improves over Visual-Only by +0.9% Top-1 and +2.2% Top-5, suggesting that shared features allow semantic cues to regularize the visual pathway. In the subject-independent setting, the gain becomes larger (+0.8% Top-1 and +2.5% Top-5), showing that semantic alignment enhances category-level consistency across subjects. Similarly, for the semantic pathway, the interaction with visual features also brings complementary benefits, confirming that partial parameter sharing is essential for mutual enhancement between the two branches.

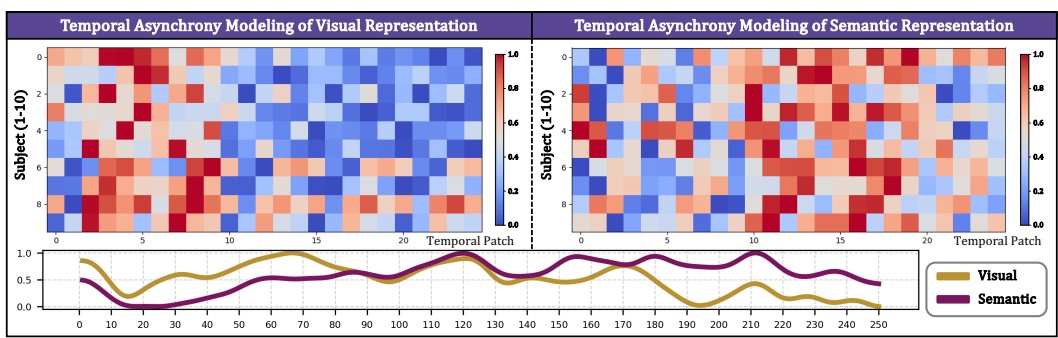

Figure 4: Visualization of temporal asynchrony: patch attention heatmap and averaged weight curve.

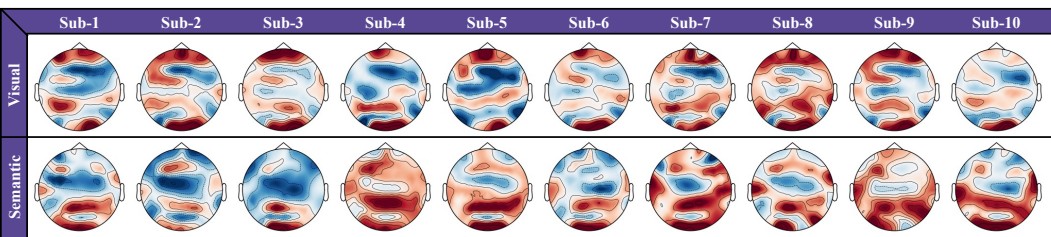

Figure 5: Visualization of spatial asynchrony: topographic map of the learnable brain network.

### 4.4.2 ABLATION ON SPATIAL-TEMPORAL ASYNCHRONY MODELING

For the Spatial–Temporal Asynchrony Modeling module, we compared three simplified configurations with the full VISTA: 1) Temporal-Only, where the spatial asynchrony modeling (i.e., brain-network learning and positional embedding before the shared spatial feature extractor) was removed; 2) Spatial-Only, where the temporal asynchrony modeling (i.e., the scalar patch weighting mechanism enhancing EEG temporal dynamics) was disabled; and 3) Both-Off, where both spatial and temporal asynchrony modeling were removed.

As shown in Table 4, all simplified variants exhibit noticeable performance drops compared to the full VISTA, confirming the complementary roles of temporal and spatial asynchrony modeling. Temporal asynchrony improves fine-grained temporal discriminability, whereas spatial asynchrony enhances spatial consistency and cross-region integration. Their joint modeling yields the highest accuracy and robustness, highlighting the necessity of incorporating both forms of asynchrony in VISTA.

### 4.5 REPRESENTATION ANALYSIS

#### 4.5.1 TEMPORAL ASYNCHRONY MODELING ANALYSIS.

Figure 4 reveals distinct temporal adaptation patterns: the visual representation (left) focuses on early temporal patches, reflecting its reliance on fine-grained object features, while the semantic representation (right) emphasizes later temporal patches, consistent with the delayed processing of high-level semantic concepts (Grill-Spector & Kanwisher, 2005). This temporal shift highlights the differential processing of visual and semantic information, where visual features are processed first, followed by abstract semantic concepts.

It is worth noting that the boundary between "visual" and "semantic" representations in EEG is not strictly separable. EEG signals inherently reflect overlapping neural activities, and partial temporal overlap between visual and semantic components may correspond to natural integration processes in the brain during short-term visual observation. Therefore, the observed asynchrony should be interpreted as a tendency rather than a rigid division, reflecting the gradual transition from perceptual to conceptual encoding within dynamic neural processing.

### 4.5.2 SPATIAL ASYNCHRONY MODELING ANALYSIS.

Figure 5 highlights spatial asynchrony in brain activity. To visualize spatial asynchrony of VISTA, we compress the learned 2D brain network into node-wise activation scores using degree centrality (DC). For each EEG electrode, its DC is computed as the sum of incoming and outgoing connection strengths: $DC(A_i) = \sum_{j=1}^{n} A(i, j) + \sum_{k=1}^{n} A(j, i) - 2A(i, i)$, where $n$ is the number of EEG channels, $A$ denotes the adjacency matrix, and $A(i, j)$ represents the directed connection strength from node $i$ to node $j$. This transforms the adjacency matrix into a 1D vector, assigning each node a single connectivity-based activation value, which is then used to render the EEG topographic map.

The visual representation (top) shows prominent activation in the prefrontal cortex, associated with eye movement and attention during object recognition. In contrast, the semantic representation (bottom) exhibits more distributed activity, suggesting broader cortical engagement for integrating semantic information (Grill-Spector & Kanwisher, 2005). This spatial divergence aligns with the hierarchical processing in the human brain, where visual details are localized, while semantics require broader cortical involvement.

### 4.5.3 VISUAL RECONSTRUCTION.

Although designed for object recognition, the intermediate representation also support high-quality visual reconstruction as shown in Figure 1.B. **More details are provided in Appendix A.6.** Besides, the representation similarity matrices across different image categories are reported in Appendix A.5.

## 5 CONCLUSION

In this study, we developed VISTA, a novel brain visual decoding framework for neural object recognition. By leveraging visual-semantic disentanglement, VISTA effectively separates EEG responses into fine-grained visual and coarse-grained semantic representations. Additionally, spatial-temporal asynchrony modeling captures dynamic interactions between these modalities, further enhancing their separability. Through alignment with CLIP embeddings, VISTA bridges the gap between EEG features and external visual and semantic spaces, enabling robust knowledge transfer. Our approach addresses the challenges posed by the heterogeneity and spatial-temporal asynchrony of EEG signals, achieving state-of-the-art zero-shot object recognition performance on large-scale EEG datasets. Experiments demonstrate its effectiveness, interpretability, and potential for real-world visual decoding applications, advancing EEG-based visual decoding and multi-modal integration.

### ETHICS STATEMENT

This work adheres to the ICLR Code of Ethics. Our study does not involve human subjects, personally identifiable information, or sensitive data, and therefore does not raise concerns regarding privacy, security, or potential harm. All datasets used are publicly available and widely adopted in the research community. We have carefully followed ethical standards in conducting experiments, reporting results, and citing related work.

### REPRODUCIBILITY STATEMENT

We have made significant efforts to ensure the reproducibility of our work. The main paper provides a detailed description of our model architecture and training procedure (Section 3). Additional implementation details, including hyperparameter settings, data preprocessing steps, and training/testing scripts, are provided in the Section 4 and supplementary materials. We also release the full source code (model implementation, training, and evaluation) as an anonymous repository to facilitate independent reproduction of our results.

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

# A APPENDIX

## A.1 LLM USAGE STATEMENT

We used a large language model (OpenAI ChatGPT, GPT-5) as a general-purpose assistant exclusively for minor language refinement, including grammar correction, style improvement, and conciseness enhancement. The LLM was not involved in research ideation, experimental design, data analysis, or result interpretation. All technical content, analyses, and conclusions were fully developed, reviewed, and verified by the authors, who take complete responsibility for the final manuscript.

## A.2 DETAILS OF THE DATASETS AND PREPROCESSING

### A.2.1 THINGS-EEG

We conducted experiments on a large-scale EEG dataset THINGS-EEG (Gifford et al., 2022) comprising recordings from ten healthy adult participants (mean age 28.5 years; 8 female, 2 male), all with normal or corrected-to-normal vision. Visual stimuli were selected from the THINGS database (Hebart et al., 2019; 2020), which includes naturalistic object images spanning 1854 distinct concepts grouped into 27 higher-level categories. For model training and evaluation, the object concepts were split into 1654 training and 200 testing categories, ensuring category balance. Ten images per concept were used for training, and one image per concept was reserved for testing.

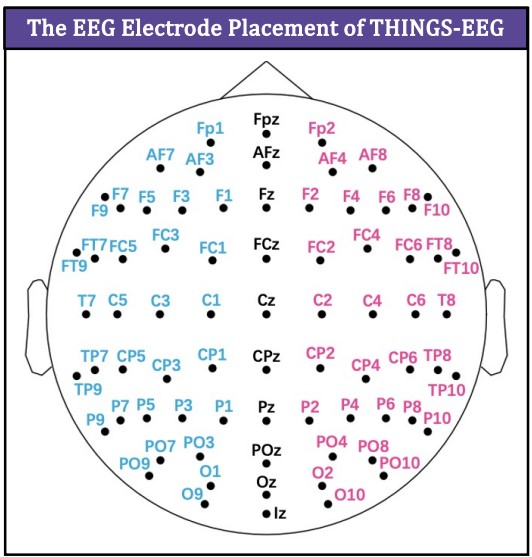

Figure 6: Electrode montage used in THINGS-EEG dataset.

Recordings followed a rapid serial visual presentation (RSVP) (Thorpe et al., 1996) paradigm with an orthogonal target detection task to sustain participant attention. Each trial consisted of 20 sequentially presented images, displayed for 100 ms each and separated by 100 ms blanks (200 ms SOA). Each subject completed four sessions, yielding 82,160 trials per participant (16,540 training trials repeated four times and 200 test images repeated 80 times). Participants maintained fixation throughout, and EEG signals were collected with a 64-channel cap arranged according to the 10–10 system.

Figure 6 shows the electrode layout. EEG was acquired at 1000 Hz, band-pass filtered between 0.1-100 Hz, and referenced to Fz. Offline processing involved epoching from -200 ms to 800 ms relative to stimulus onset, baseline correction, and downsampling to 100 Hz. Target trials were discarded, and multivariate noise normalization was applied per session without further artifact rejection.

For analysis, EEG trials were segmented from 0–1000 ms after stimulus onset, and baseline-corrected using the mean pre-stimulus voltage (-200-0 ms). All 64 channels were retained, and signals were resampled to 250 Hz for computational efficiency. Repeated presentations of the same image were

averaged to reduce noise and stabilize representations. Image stimuli were resized to 224×224 and normalized prior to encoding.

### A.2.2 THINGS-MEG

We additionally evaluated our model on the THINGS-MEG dataset (Hebart et al., 2023), which provides magnetoencephalography (MEG) recordings from four participants across 12 sessions. Visual stimuli were drawn from the THINGS database (Hebart et al., 2019), encompassing 1854 object concepts and over 26,000 curated naturalistic images. During each session, participants viewed images for 500 ms followed by a blank interval of 1000 ± 200 ms while maintaining central fixation. An orthogonal oddball detection task was employed to ensure sustained attention.

For zero-shot evaluation, 200 object concepts were excluded from training. The training set comprised 1854 concepts with 12 unique images per concept (1854×12×1), and the test set included 200 concepts with one image repeated 12 times (200×1×12). MEG data were recorded with a 271-channel whole-head system, epoched from 0–1000 ms post-stimulus, filtered between 0.1–100 Hz, and downsampled to 200 Hz. Repeated trials for each image were averaged to enhance signal-to-noise ratio. Due to the limited number of subjects, statistical analyses were not performed, and this dataset was used solely as supplementary validation for EEG results.

### A.3 DETAILS OF THE EXPERIMENTAL SETTING

#### A.3.1 EXPERIMENTAL SETTING

For evaluation, we conducted object recognition experiments in subject-dependent and subject-independent settings, consistency with NICE (Song et al., 2024). For each training, we randomly selected 740 trials from the training set to serve as the validation set. During training, the model was saved when the validation loss plateaued, and the best-performing model on the validation set was retained. After training, we performed a single evaluation on the test set to compute the object recognition accuracy. To mitigate the impact of randomness in validation set selection, we repeated each experiment five times and reported the average performance across all runs. This ensures the robustness and reliability of the experimental results for all compared methods.

#### A.3.2 COMPARISON METHODS

We compared VISTA with several recent EEG decoding approaches, and below are the details of comparison methods:

- **DGCNN [IEEE TAFFC'2018] (Song et al., 2018):** The Dynamical Graph CNN employs a learnable adjacency matrix along with Chebyshev filters to capture dynamic relationships for emotion classification.
- **EEGNet [JNE'2018] (Lawhern et al., 2018):** A lightweight convolutional network that efficiently extracts EEG temporal-spatial features, optimized for BCI applications.
- **EEG-Conformer [IEEE TNSRE'2022] (Song et al., 2022):** This architecture combines convolutional neural networks with Transformer modules for feature extraction in sequential EEG data.
- **EEG-ChannelNet [IEEE TPAMI'2020] (Palazzo et al., 2020):** EEG-ChannelNet learns a brain manifold from EEG data and aligns it with visual features via a siamese network, enabling visual information decoding and improving image classification and saliency detection.
- **Mb2C [ACM MM'2024] (Wei et al., 2024):** The Multimodal Bidirectional Cycle Consistency framework utilizes dual-GAN architectures to generate and reconcile modality-specific features, effectively bridging the modality gap in EEG brain decoding.
- **NICE [ICLR'2024] (Song et al., 2024):** A self-supervised framework that learns image representations from EEG by incorporating attention modules to capture spatial correlations within brain activity.
- **ATM [NeurIPS'2024] (Li et al., 2024):** The Adaptive Thinking Mapper projects neural signals into a shared subspace, enabling zero-shot visual decoding.

- **CognitionCapturer** [AAAI'2025] **(Zhang et al., 2024):** This unified framework enhances EEG representations by leveraging multimodal data including image depth and modality-specific expert encoders.

- **EEG-CLIP** [Neural Networks'2025] **(Cao et al., 2025):** A Transformer-based framework that maps EEG signals into CLIP's visual space and refines them via a diffusion prior. This design enables accurate, high-fidelity decoding of visual perception from neural activity.

- **BrainFLORA** [ACM MM'2025] **(Li et al., 2025b):** A unified framework that integrates EEG, MEG, and fMRI through multimodal large language models with modality-specific adapters. This design constructs a shared neural representation across modalities, enabling cross-subject visual retrieval and revealing consistent concept alignment in the brain.

- **SRT** [ICCV'2025] **(Kim et al., 2025):** A retrieval-augmented generation framework that decodes visual perception from EEG signals. It aligns EEG with image–text embeddings to retrieve semantically related samples, which guide a diffusion model to generate high-fidelity visual reconstructions.

### A.3.3 Reliability of Experimental Results

**All comparisons are fair and use identical splits and evaluation as ATM (Song et al., 2024) and NICE (Li et al., 2024).** Since the test set for each subject contains 200 samples, and the validation set is randomly selected, this may cause some fluctuations in the results. To ensure reliability, the final results of all experiments are reported as the average over 5 trials. **Code is included in supplementary materials and will be released publicly upon acceptance.**

### A.4 Additional Parameter Analysis

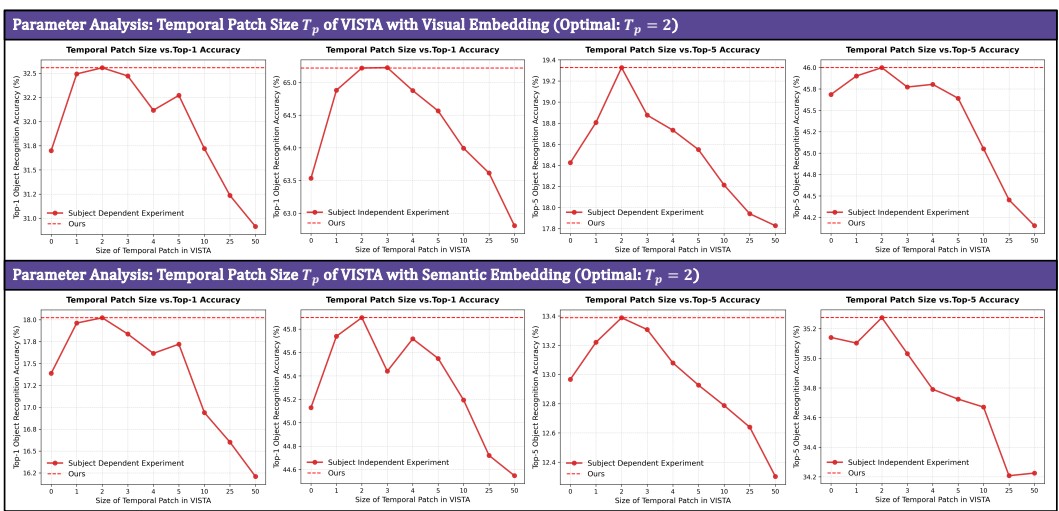

Figure 7: Parameter analysis of temporal patch size $T_p$.

### A.4.1 Temporal Patch Size $T_p$

As shown in Figure 7, we conducted a detailed analysis on the impact of temporal patch size $T_p$. Experiments were performed under both subject-dependent and subject-independent settings, evaluating Top-1 and Top-5 zero-shot image retrieval accuracy. We visualized results for both visual and semantic embeddings. The findings indicate that finer patch segmentation consistently improves recognition performance, whereas overly coarse patches tend to degrade feature quality. Compared to the no-patch baseline ($T_p = 0$), fine-grained patches yield a notable performance boost. Among all settings, $T_p = 2$ provides the most stable and optimal results across conditions.

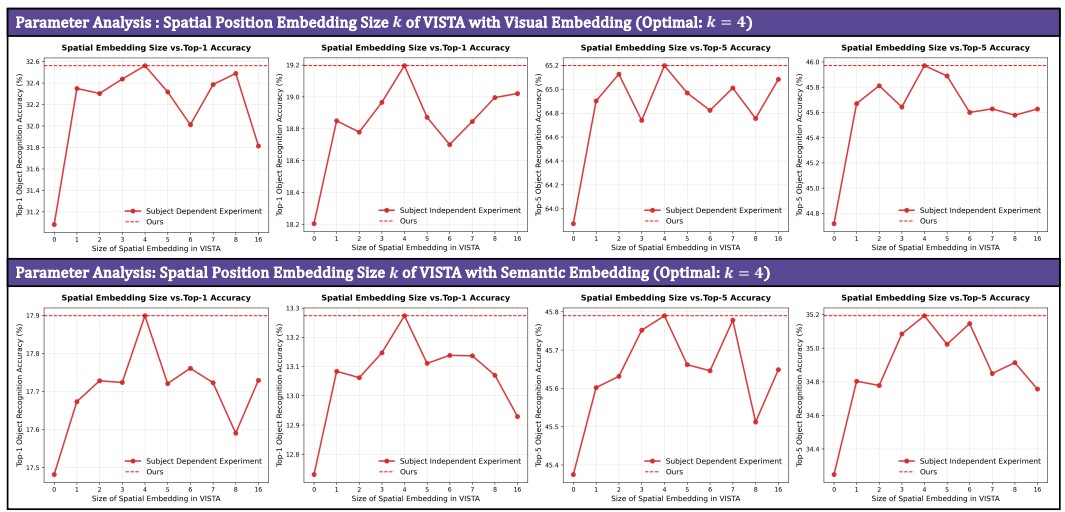

Figure 8: Parameter analysis of spatial position embedding size $k$.

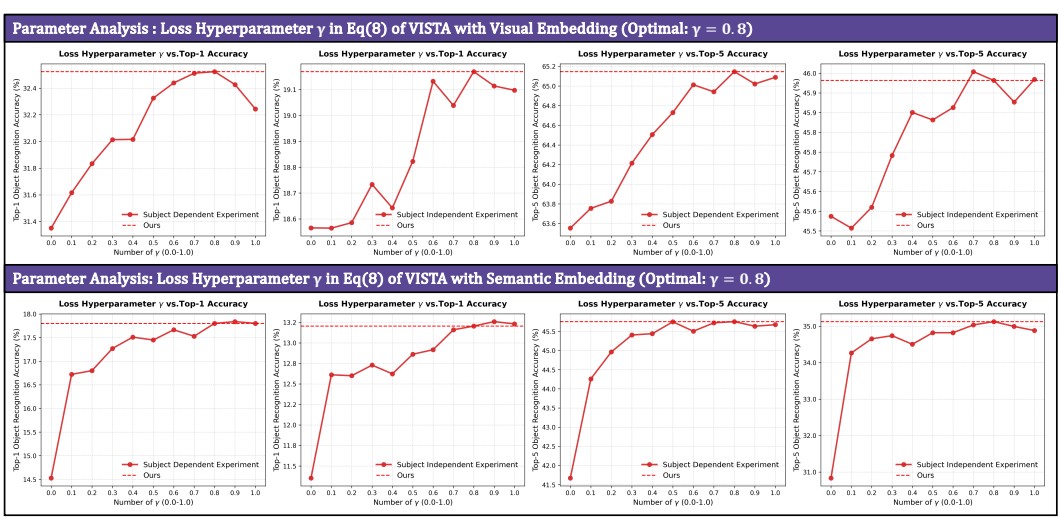

Figure 9: Parameter analysis of loss weight $\gamma$.

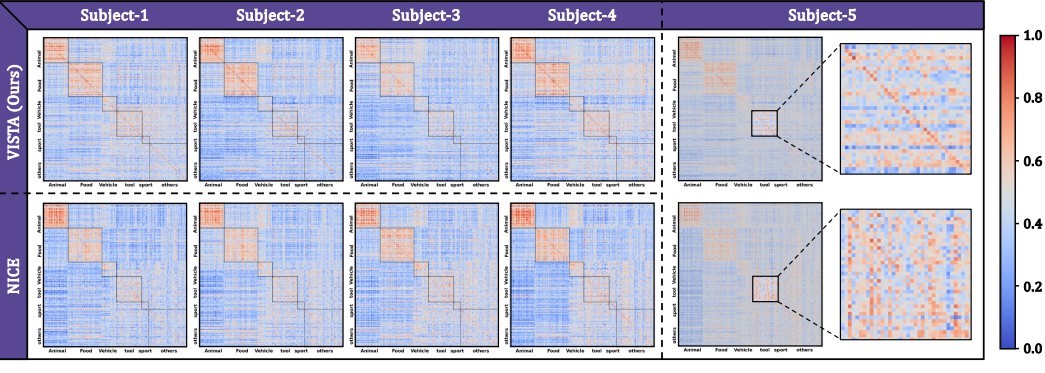

Figure 10: Representational similarity matrix (RSM) comparison of VISTA, and NICE on the THINGS-EEG dataset.

### A.4.2 PATIAL POSITION EMBEDDING SIZE $k$

As shown in Figure 8, we further investigated the effect of the position embedding size $k$ in spatial modeling. Similar to the previous analysis, we report Top-1 and Top-5 zero-shot image retrieval accuracy under both subject-dependent and subject-independent settings, using both visual and semantic embeddings. The results show that position embeddings derived from low-rank decomposition of brain graphs significantly enhance recognition accuracy. Among all configurations, $k = 4$ achieves the best trade-off between stability and performance. Larger $k$ values may lead to marginal gains but introduce increased computational overhead.

### A.4.3 HYPERPARAMETER SEMANTIC LOSS WEIGHT $\lambda$

As shown in Figure 9, we analyzed the influence of the loss weight $\lambda$, which controls the contribution of the semantic embedding in the joint alignment objective. Values of $\lambda \in [0, 1]$ were tested, and performance was evaluated under both subject-dependent and subject-independent settings using Top-1 and Top-5 metrics. Results indicate that introducing semantic supervision improves the accuracy of both visual and semantic embeddings. In particular, $\lambda = 0.8$ achieves the most stable performance. Higher values of $\lambda$ may overemphasize semantic guidance and slightly degrade visual embedding accuracy.

## A.5 ADDITIONAL REPRESENTATION ANALYSIS

### A.5.1 REPRESENTATIONAL SIMILARITY MATRICES

To evaluate the representational consistency of VISTA, we visualize the representational similarity matrix (RSM) as a heatmap using data from the first five subjects in the THINGS-EEG dataset. The RSM quantifies the similarity of neural representations across different conditions.

We compare VISTA with the state-of-the-art baselines NICE (Song et al., 2024), which is commonly used for EEG visual decoding. As shown in Figure 10, VISTA demonstrates a more distinct and well-structured pattern along the main diagonal of the matrix compared to NICE, indicating that VISTA effectively captures and maintains representational integrity across different EEG trials. To further highlight this improvement, we provide a zoomed-in view of a selected region in Figure 10. The magnified section reveals finer details of the RSM, showing that VISTA produces clearer and more pronounced similarity patterns than the baselines. This suggests that VISTA learns more discriminative and stable representations, which is crucial for both object recognition and image reconstruction.

### A.5.2 VISUALIZING VISTA'S ZERO-SHOT IMAGE RETRIEVAL

Due to space limitations, we visualize the object recognition retrieval results for the first 12 test samples from the 200-sample test set of the THINGS-EEG dataset. As shown in Figure 11, VISTA retrieves the correct object within Top-5 predictions for most samples. Even when the correct target is not ranked as Top-1, the model tends to retrieve highly relevant alternatives, demonstrating its strong ability to capture object semantics and category-level relationships. For example, in case 1 (aircraft carrier), although the correct retrieval appears at Top-3, Top-1 to Top-5 results (ferry, cruise ship, submarine, and boat) are all semantically related to the target object. This suggests that VISTA effectively recognizes the structural and contextual features of the object, even if the exact match is not ranked first. Similarly, in case 2 (antelope), all retrieved results belong to the broader animal category, indicating that VISTA can generalize across visually similar concepts, preserving category-level consistency.

However, some challenges remain. In case 5 (banana), the correct retrieval only appears at Top-8, while the preceding results are mostly irrelevant objects. This suggests difficulty in prioritizing fine-grained features early in the rankings. Similarly, in case 7 (basil), while the model identifies green leafy features, it confuses basil with visually similar plants, such as strawberry and lettuce. Such misclassifications may stem from EEG signal variability, subject attention shifts, or inherent limitations in distinguishing subtle differences between fine-grained categories.

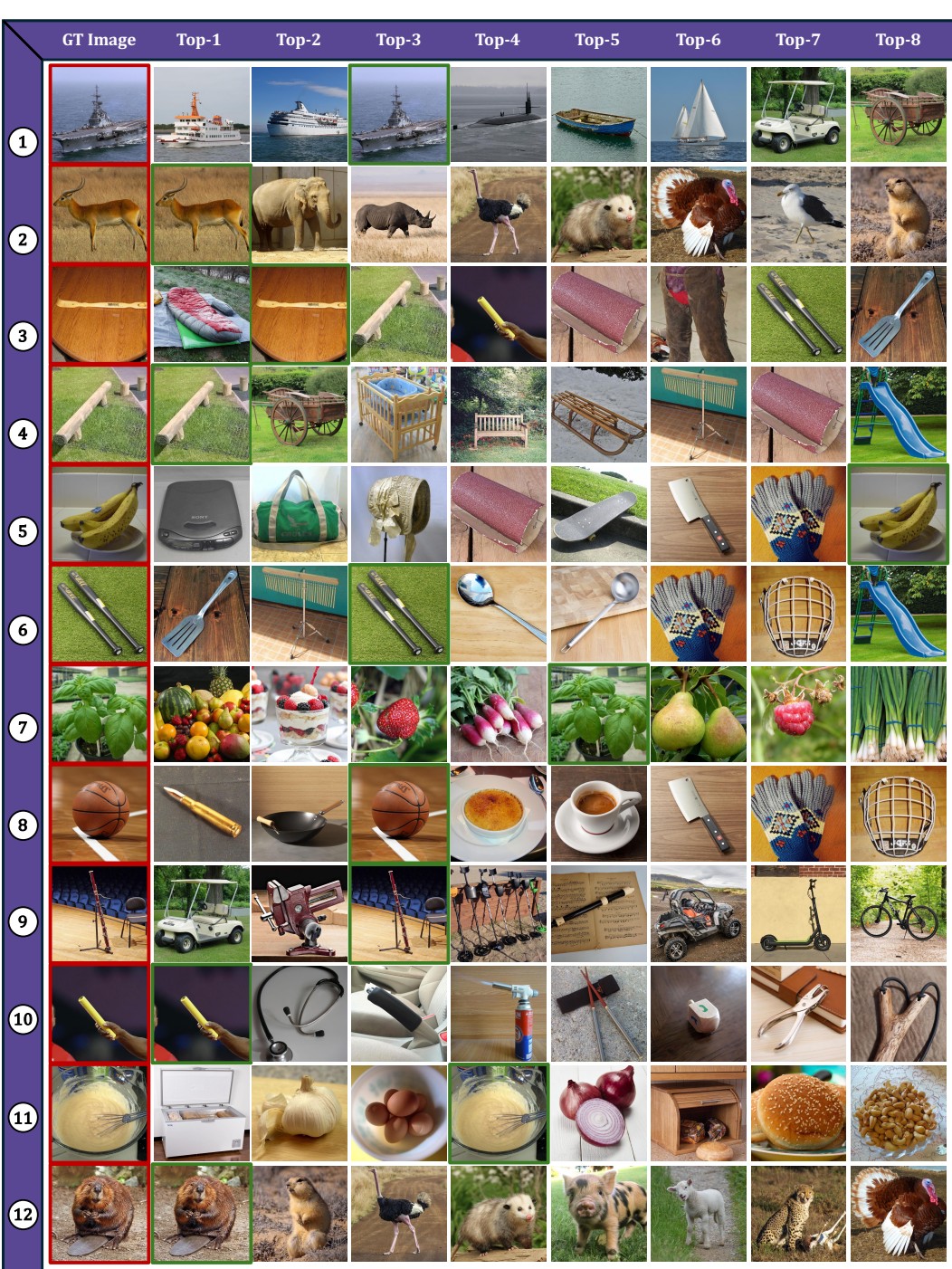

Figure 11: Retrieval results of VISTA in the 200-way zero-shot object recognition task on the THINGS-EEG dataset. (Red: ground truth image; green: retrieval right image.)

Despite these occasional errors, VISTA demonstrates strong object recognition capabilities, with most correct predictions appearing within the Top-5. Future improvements could further enhance fine-grained feature recognition and reduce errors in visually similar categories.

### A.6 VISUALIZING VISTA'S IMAGE RECONSTRUCTION

#### A.6.1 STABLE DIFFUSION XL AND IP-ADAPTER.

To enable high-quality image reconstruction from EEG signals, we leverage Stable Diffusion XL (SDXL) (Podell et al., 2023), a generative model known for its ability to produce photorealistic and semantically rich images. To bridge the modality gap between neural signals and visual content, we incorporate the IP-Adapter (Ye et al., 2023) to get better input representation of diffusion model. Specifically, we use CLIP-ViT-H/14[3] (Radford et al., 2021) to encode EEG representations, enabling them to effectively influence the denoising path within the SDXL U-Net, thus guiding the model toward more precise and semantically coherent outputs. For faster inference, we adopt SDXL-Turbo[4], an optimized lightweight variant of SDXL that delivers comparable visual fidelity with substantially reduced generation latency, making it well-suited for real-time neural decoding scenarios.

#### A.6.2 IMAGE RECONSTRUCTION PERFORMANCE.

We provide additional qualitative results of VISTA on the THINGS-EEG dataset in Figure 12, Figure 13, and Figure 14. These visualizations highlight reconstruction outcomes for three representative semantic categories: vehicles, food, and animals.

Overall, VISTA produces reconstructions that are semantically faithful to the original stimuli, yet distinct error patterns emerge across categories. For vehicles, reconstruction quality is generally high: global structure and shape (e.g., cars, airplanes, boats) are often preserved. However, fine-grained discrepancies occasionally arise, such as bicycles appearing as multiple overlapping objects or additional human figures being introduced. These errors indicate that while object-level semantics are well captured, spatial precision remains imperfect.

Among the three categories, food items show the most accurate reconstructions. Generated images consistently retain key textures, shapes, and colors, producing clear and coherent representations with minimal artifacts. This suggests that EEG responses to food-related stimuli may be more distinct and easier for the model to decode.

In contrast, animals exhibit the highest reconstruction variability. While certain features such as fur texture or body contours are captured, some outputs show semantically divergent results—for example, a lamb being reconstructed as an elderly human figure with white hair and clothing, or cats rendered as brown-clothed individuals with curly hair. These errors may reflect associative activations or attentional confounds during EEG recording, highlighting the inherent challenge of decoding fine-grained animal representations from noisy neural signals.

These findings indicate that VISTA effectively recovers coarse object semantics across categories, yet future work should focus on improving spatial fidelity and reducing category-specific misalignment, particularly for visually complex or perceptually overlapping classes such as animals.

Table 5: Quantitative assessments of reconstruction quality on the THINGS-EEG dataset.

| Method | SSIM↑ | Alex(2)↑ | Alex(5)↑ | Incep↑ | CLIP↑ | EffNet-B↓ | SwAV↓ |
|---|---|---|---|---|---|---|---|
| NICE [ICLR'24] | 0.353 | 0.774 | 0.87 | 0.736 | 0.789 | 0.825 | 0.588 |
| ATM [NeurIPS'2024] | 0.351 | 0.779 | 0.873 | 0.745 | 0.792 | 0.823 | 0.591 |
| CognitionCapturer [AAAI'2025] | 0.346 | 0.768 | 0.866 | 0.737 | 0.781 | 0.832 | 0.595 |
| **VISTA** [Ours] | **0.355** | **0.789** | **0.882** | **0.746** | **0.798** | **0.813** | **0.585** |

---

[3]https://huggingface.co/laion/CLIP-ViT-H-14-laion2B-s32B-b79K
[4]https://huggingface.co/stabilityai/sdxl-turbo

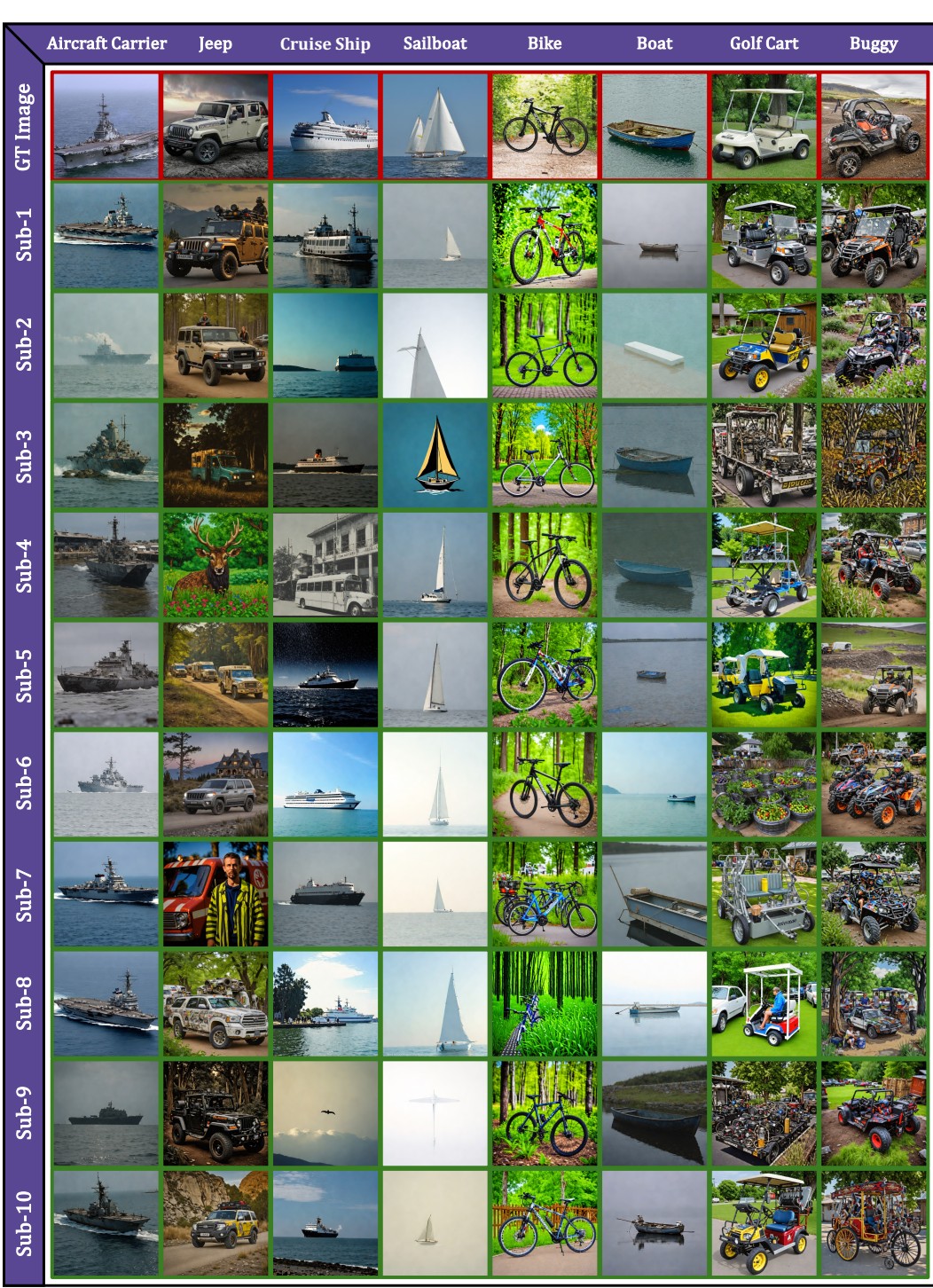

Figure 12: Reconstructed images of vehicles using VISTA. (Red: ground truth image; green: reconstructed image.)

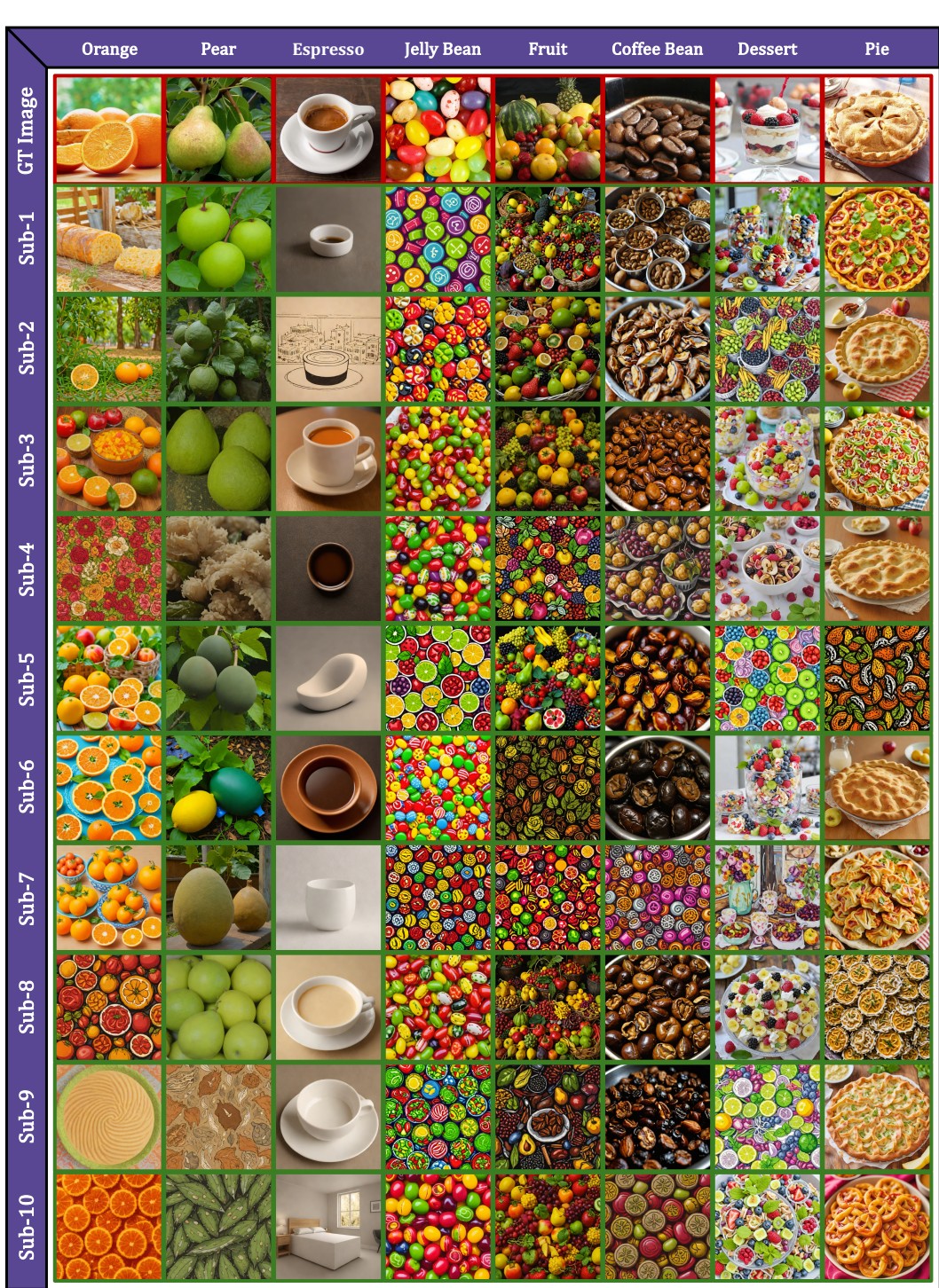

Figure 13: Reconstructed images of food items using VISTA. (Red: ground truth image; green: reconstructed image.)

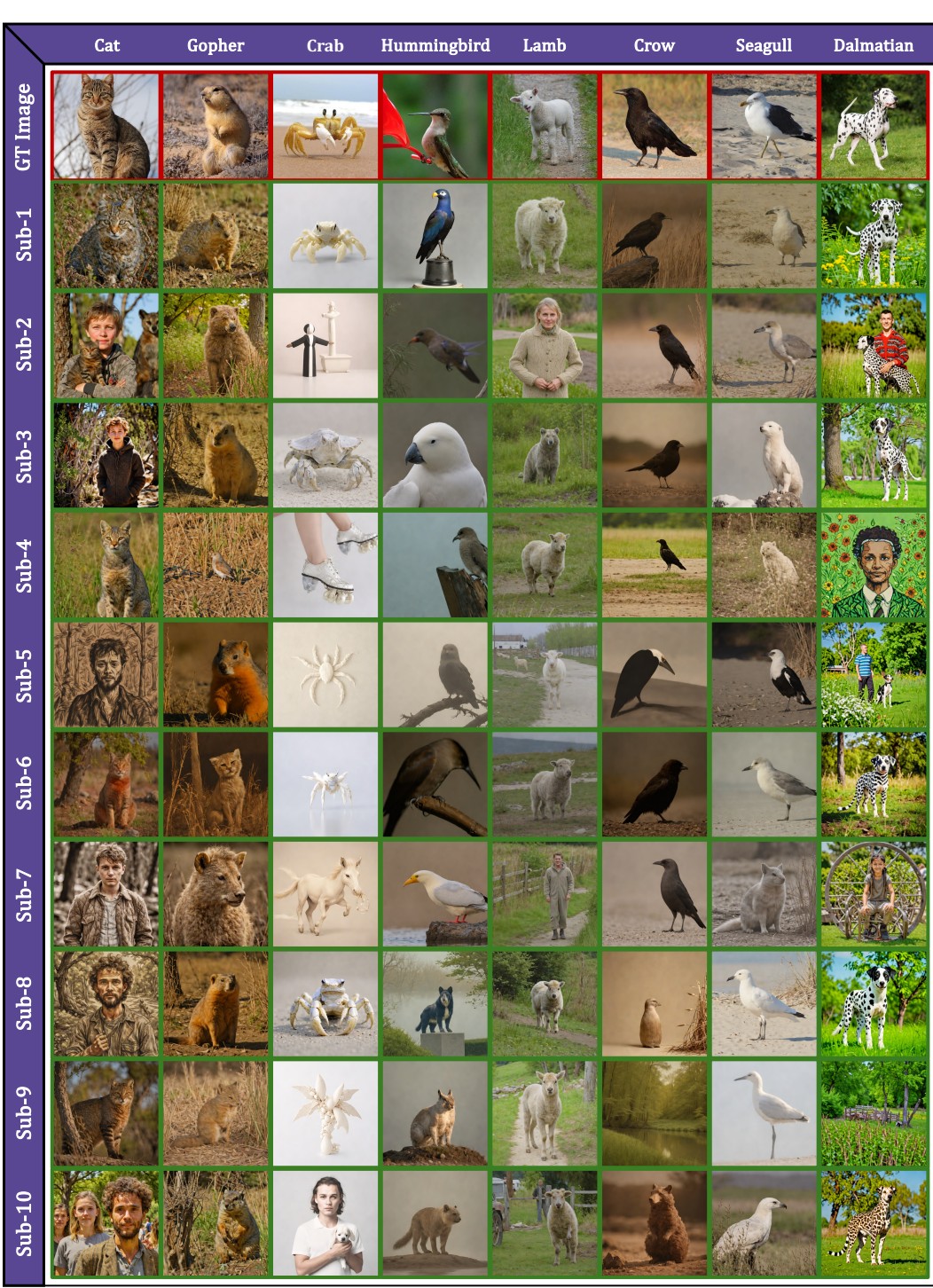

Figure 14: Reconstructed images of animals items using VISTA. (Red: ground truth image; green: reconstructed image.)

### A.6.3 QUANTITATIVE METRICS OF RECONSTRUCTED IMAGE.

We now provide quantitative evaluation of EEG-to-image reconstruction, including standard metrics used in prior work:

- **SSIM:** measures low-level structural similarity;

- **AlexNet (2/5) and Inception:** assess mid/high-level perceptual feature similarity;

- **CLIP:** evaluates alignment in joint vision-language space, relevant for semantic fidelity;

- **EffNet and SwAV:** capture feature compactness and unsupervised visual consistency.

As shown in Table 5, VISTA achieves consistently better scores, which suggests that our EEG encoder contributes to more semantically faithful reconstructions.

### A.7 MODEL COMPUTING RESOURCE COMPARISON

Table 6: Comprehensive comparison of computational requirements and efficiency.

| Methods | FLOPs. | Param. | Train time (s) | Test time (s) |
|---|---|---|---|---|
| NICE [ICLR'24] | 42.7354 M | 2630.940 K | 0.0412 | 0.0058 |
| ATM [NeurIPS'2024] | 98.209 M | 3072.662 K | 0.0524 | 0.0088 |
| CognitionCapturer [AAAI'2025] | 178.973 M | 2954.073 K | 0.0591 | 0.0056 |
| **VISTA** [Ours] | 89.865 M | 3034.576 K | 0.0505 | 0.0063 |

Table 6 summarizes the computational cost, parameter size, and processing efficiency of VISTA compared to existing baselines. Three observations can be drawn:

**Efficiency-Accuracy Trade-off**: While CopCap incurs the highest computational cost (179.0M FLOPs), VISTA operates with significantly fewer computations (89.9M FLOPs), achieving a favorable trade-off between complexity and efficiency. Compared to ATM, VISTA reduces FLOPs by nearly 10%, while offering improved runtime performance (6.3 ms vs. 8.8 ms per trial).

**Compact and Effective Parameter Design**: VISTA employs 2.83M parameters, fewer than both ATM (3.07M) and CopCap (2.95M), indicating a streamlined architecture. Despite this, it maintains strong performance, suggesting its visual-semantic disentanglement and spatial-temporal modeling strategy make more efficient use of network capacity.

**Practical Deployment Readiness**: In terms of training and inference speed, VISTA maintains low latency (0.0505 s train / 0.0063 s test), comparable to lightweight baselines like NICE, while offering a much richer representational capacity. Its compact design and real-time inference speed make it well-suited for practical brain decoding scenarios.

### A.8 SUPPLEMENTARY CROSS-TASK VALIDATION: EXPERIMENTS ON EEG CLASSIFICATION

While our primary objective is zero-shot EEG-to-image retrieval rather than EEG classification, we are also interested in understanding whether VISTA's feature extractor generalizes to other EEG tasks. Therefore, we additionally report the performance of the VISTA encoder on several widely used EEG classification datasets, including ImageNet-EEG (Spampinato et al., 2017; Kavasidis et al., 2017), DEAP (Koelstra et al., 2011), SEED (Zheng & Lu, 2015) and SEED-IV (Zheng et al., 2018).

It is important to emphasize that all these datasets target closed-set classification, whereas our main task is open-set zero-shot retrieval without any classifier. Thus, for fairness, we only attach a simple classifier on top of the frozen VISTA encoder to extract task-specific features. These results should not be interpreted as required evaluation metrics, but rather as supplementary evidence of representation quality.

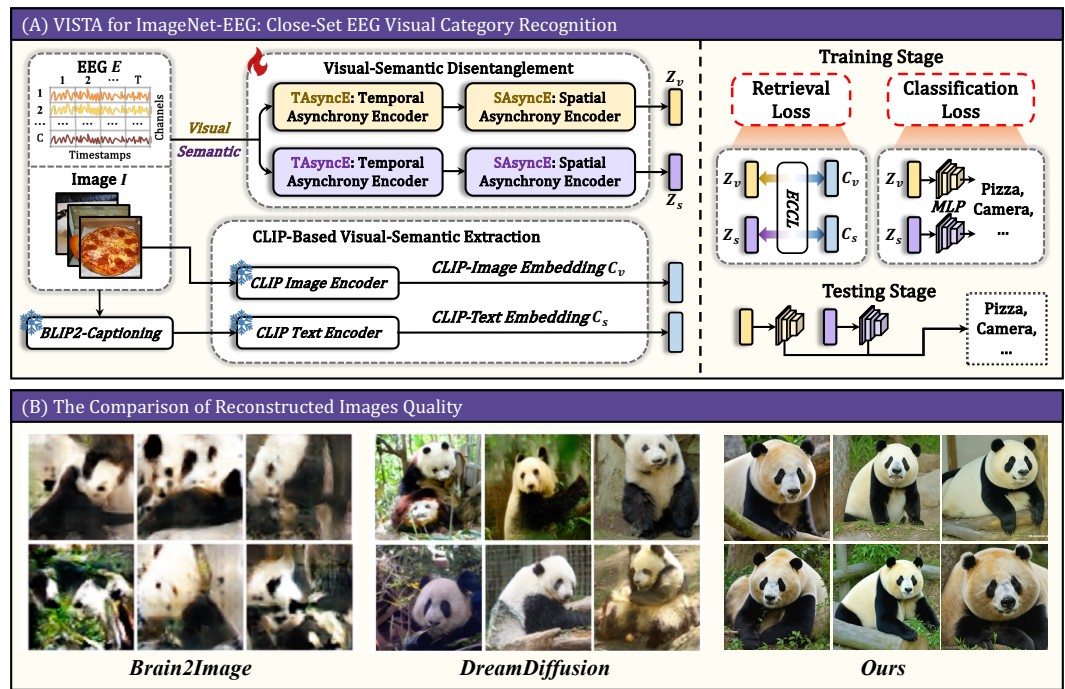

Figure 15: (A) The diagram VISTA framework for close-set EEG visual category recognition task. (B) The comparison of reconstructed images quality (Panda) of the ImageNet-EEG dataset.

### A.8.1 IMAGENET-EEG: CLOSE-SET EEG VISUAL CATEGORY RECOGNITION

**ImageNet-EEG** dataset is designed for closed-set EEG-based visual category classification. It contains 40 distinct ImageNet-derived classes, each represented by 50 images, resulting in 2,000 visual stimuli presented to six subjects. EEG was recorded using a 128-channel system, yielding approximately 11,000 trials. Following prior work, we adopt an 8:1:1 split for training, validation, and testing. For consistency, we use the same visual encoder as in THINGS-EEG experiments, i.e., `CLIP-ViT-H-14-laion2B-s32B-b79K`. Other experimental details follow BrainVis (Fu et al., 2025), and the flow diagram is in Figure 15 (A).

Table 7: Image object classification performance (Mean/Std%) of VISTA and comparison methods in subject-dependent setting on the ImageNet-EEG dataset.

| Method | Top-1 Acc | Top-3 Acc | Top-5 Acc | F1-Score |
|---|---|---|---|---|
| Brain2Image (Kavasidis et al., 2017) | 16.2 / - | 37.9 / - | 55.2 / - | 16.1 / - |
| BrainVis (Fu et al., 2025) | 47.7 / - | 75.2 / - | 91.2 / - | 43.2 / - |
| KD-STFT (Ferrante et al., 2024) | 41.2 / 11.3 | 75.3 / 10.7 | 87.8 / 8.1 | 40.3 / 11.6 |
| **VISTA (Visual)** | **65.5 / 10.2** | **91.2 / 02.6** | **97.8 / 01.4** | **61.7 / 11.5** |
| **VISTA (Semantic)** | 61.2 / 13.5 | 87.0 / 05.1 | 93.5 / 03.0 | 60.3 / 13.8 |

As we can seen in Table 7, VISTA significantly outperforms prior methods, demonstrating the effectiveness of disentangling visual and semantic embeddings. The visual branch achieves slightly higher Top-1, Top-3, and Top-5 accuracy than the semantic branch, which is consistent with the fact that ImageNet-EEG focuses on visual object categorization. These results further confirm that our EEG encoder captures robust and discriminative visual features that generalize across subjects.

In addition, Figure 15 (B) compares the reconstructed images (category: Panda) across all six subjects, with Brain2Image (Kavasidis et al., 2017) and DreamDiffusion (Bai et al., 2024) as baselines, which further highlight the superiority of VISTA. While Brain2Image produces blurry and structurally distorted outputs due to its GAN-based decoder, and DreamDiffusion often fails to preserve category-

specific details, VISTA generates reconstructions with sharper contours, richer textures, and better alignment to the target semantics, validating the advantage of disentangled visual representation learning for EEG visual decoding.

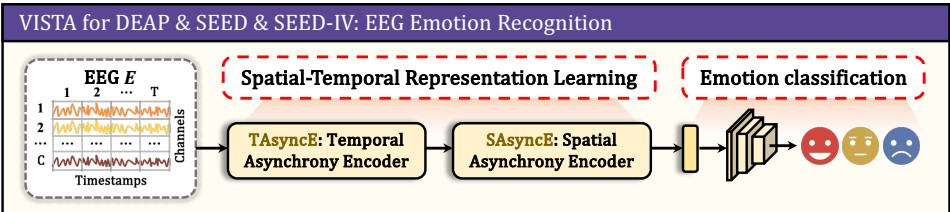

Figure 16: The diagram VISTA framework for emotion recognition task.

### A.8.2 DEAP: EEG EMOTION RECOGNITION

We further evaluate on the **DEAP** dataset, designed for emotion analysis. It contains EEG from 32 subjects watching 40 one-minute affective video clips, each rated on arousal, valence, and dominance. Since the dataset focuses on emotional states rather than visual or semantic processing, we use the VISTA encoder as a general-purpose feature extractor. Experimental details follow TSception (Ding et al., 2023). Figure 16 illustrates how to use VISTA's spatial-temporal encoder to extract emotional representations for emotion recognition.

Table 8: Emotion Recognition performance (Mean/Std%) of VISTA and comparison methods in subject-dependent setting on the DEAP dataset.

| Method | Valence Acc | Valence F1 | Arousal Acc | Arousal F1 |
|---|---|---|---|---|
| DBN (Zheng et al., 2014) | 86.23 / 07.66 | 87.05 / 06.91 | 85.20 / 07.13 | 86.62 / 06.84 |
| EEGNet (Lawhern et al., 2018) | 92.24 / 05.97 | 92.72 / 05.39 | 93.86 / 05.21 | 94.38 / 04.46 |
| DGCNN (Song et al., 2018) | 93.12 / 08.05 | 94.15 / 06.77 | 92.94 / 06.75 | 93.40 / 06.17 |
| EEG-ConFormer (Song et al., 2022) | 93.36 / 03.89 | 93.65 / 04.50 | 94.18 / 05.14 | 94.52 / 04.85 |
| TSception (Ding et al., 2023) | 93.18 / 04.15 | 93.40 / 04.66 | 93.74 / 04.96 | 94.31 / 05.22 |
| BSTT (Liu & Jia, 2023) | 92.71 / 04.66 | 93.06 / 04.11 | 92.45 / 05.22 | 93.69 / 05.40 |
| LResCapsule (Fan et al., 2024) | 95.15 / 03.51 | - | 95.77 / 03.82 | - |
| FAT (Wang et al., 2025) | 90.10 / 02.71 | - | 89.18 / 03.50 | - |
| **VISTA** | **95.47 / 03.62** | **95.08 / 04.09** | **95.96 / 04.02** | **95.14 / 04.71** |

As we can seen in Table 8, VISTA achieves the highest accuracy and F1 scores across both valence and arousal dimensions, indicating that its learned EEG representations are transferable beyond visual decoding to emotion recognition. This suggests that VISTA's spatial-temporal encoder also can captures generalizable emotion representation from EEG signal.

### A.8.3 SEED AND SEED-IV: EEG EMOTION RECOGNITION

We also evaluate on the **SEED** and **SEED-IV** datasets. **SEED** records 62-channel EEG from 15 subjects across three sessions while watching positive, neutral, and negative emotion-inducing videos. **SEED-IV** records four emotional states (happy, sad, fear, neutral) from the same subjects. We use differential entropy (DE) features and standard preprocessing, and other experimental details follow DGCNN (Song et al., 2018).

As we can seen in Table 9, VISTA performs the best on the SEED dataset, and is close to the best-performing baseline (STRFLNet) on the SEED-IV dataset, highlighting its ability to extract discriminative EEG features for multi-class emotion recognition.

Table 9: Emotion Recognition performance (Mean/Std%) of VISTA and comparison methods in subject-dependent setting on the SEED and SEED-IV datasets.

| Method | SEED Acc | SEED-IV Acc |
|---|---|---|
| DBN (Zheng et al., 2014) | 86.08 / 08.34 | 66.77 / 07.38 |
| DGCNN (Song et al., 2018) | 90.40 / 08.49 | 69.88 / 16.29 |
| RGNN (Zhong et al., 2020) | 91.79 / 06.72 | 79.37 / 10.54 |
| SST-EmotionNet (Jia et al., 2020) | 96.02 / 01.86 | 84.92 / 06.66 |
| BiHDM (Li et al., 2020) | 93.12 / 06.06 | 74.35 / 14.09 |
| PGCN (Jin et al., 2024) | 96.93 / 05.11 | 82.24 / 14.85 |
| STRFLNet (Hu et al., 2025) | 96.42 / - | **92.23 / -** |
| FAT (Wang et al., 2025) | 92.10 / 06.46 | 82.30 / 13.21 |
| CLIER (Li et al., 2025a) | 93.4 / 06.31 | 90.02 / 10.37 |
| UMDA-DDSTG (Gao et al., 2025) | 95.89 / 05.78 | 84.48 / 10.15 |
| **VISTA** | **97.13 / 03.49** | 91.85 / 10.43 |

