# OpenReview forum: "VISTA: Visual-Semantic Disentanglement and Dynamic Spatial-Temporal Asynchrony for Brain Decoding"
_ICLR.cc/2026/Conference — Submitted to ICLR 2026_

### Official Review · Reviewer_zxT2 · 2025-10-23

**Soundness:** 3
**Presentation:** 3
**Contribution:** 2
**Rating:** 6
**Confidence:** 2

**Summary:**

VISTA presents a novel EEG-based neural decoding framework that disentangles visual and semantic representations of brain activity while modeling their inherent spatial-temporal asynchrony. The model divides EEG signals into time patches to capture asynchronous temporal activations and learns separate brain connectivity graphs for visual and semantic modalities via low-rank and Laplacian spectral decomposition. These representations are then aligned with CLIP’s image and text embeddings through contrastive learning to leverage large-scale visual-semantic priors. Experiments on THINGS-EEG and THINGS-MEG datasets show that VISTA achieves state-of-the-art zero-shot object recognition accuracy.

**Strengths:**

1. VISTA explicitly models asynchronous neural dynamics by separating EEG into time-specific and spatially distinct pathways for visual and semantic features, providing biologically plausible and interpretable representations.
2. By aligning EEG-derived embeddings with CLIP’s image and text spaces, the model leverages pre-trained multimodal representations for zero-shot decoding, enhancing generalization to unseen categories.
3. The framework achieves consistent state-of-the-art performance across EEG and MEG datasets.

**Weaknesses:**

1. The multi-branch architecture involving spatial-temporal encoders, graph learning, and CLIP alignment increases training overhead and may limit practical scalability for real-time brain–computer interfaces. Also, such an over-complex architecture seems lack principal novelty and application potential
2. The reliance on CLIP embeddings ties performance to the representational quality and domain coverage of CLIP, potentially restricting generalization to novel or non-visual cognitive tasks.

**Questions:**

1. How well do the learned visual and semantic brain networks correspond to actual cortical pathways known from neuroscience, and can this be quantitatively validated?
2. Could VISTA’s disentanglement and alignment approach extend to other cognitive decoding tasks (e.g., language comprehension or emotion recognition)?

---

> ### Author Response · Authors · 2025-11-15
> **Response to reviewr zxT2 (1/3)**
>
> Thanks a lot for your expertise and thoughtful comments!
>
> We address each concern below, providing both high-level clarification and detailed explanation. All the revisions have been marked in blue throughout the text.
>
>
>
> **To Weakness:**
>
> **[Weakness 1] Model Complexity and Practicality:** We acknowledge that multi-branch architectures can introduce additional computational cost. However, in VISTA, **feature extractors are partially shared between branches**, which reduces overhead. As shown in APPENDIX Table 6, VISTA’s computational requirements remain **comparable to or lower than other SOTA methods**, and careful optimization further mitigate training and inference cost.
>
> Beyond efficiency, the architecture provides **biologically interpretable visual-semantic representations**, which are crucial for neuroscience-inspired EEG decoding. In our view, this trade-off between slight computational overhead and interpretability is justified, especially in offline or research-oriented decoding scenarios.
>
> **[Weakness 2] Reliance on CLIP Embeddings:** We agree that VISTA’s current performance depends on the quality and coverage of CLIP embeddings, which is a common limitation in EEG visual decoding research. We use CLIP embeddings because:
>
> 1. CLIP provides high-quality, large-scale visual-semantic features learned from millions of image-text pairs.
> 2. THINGS-EEG alone does not contain enough images to train high-quality embeddings independently.
>
> Thus, the reliance on CLIP **is a consequence of limited dataset size rather than a limitation of VISTA itself**. If a sufficiently large EEG–image dataset were available, VISTA could learn high-quality embeddings directly from the data without relying on CLIP. This principle applies not only to image decoding, but also to other modalities such as video or auditory decoding: the reason researchers currently rely on pretrained embeddings (e.g., CLIP for images, pretrained audio embeddings for sound) is that the corresponding EEG datasets are too small (have small scale of decoding modality like image, sound and others) to allow independent training.
>
> Even so, for non-visual decoding tasks, researchers can still relatively easily find task-appropriate pretrained models in place of CLIP. For example, Neuro-3D [R1] slices 3D objects into multi-view images and feeds them into CLIP to obtain 3D-aware visual embeddings, while EEG2Video [R2] extracts video subtitles, converts them into CLIP text embeddings, and then uses these embeddings to guide video reconstruction. In this sense, using CLIP in VISTA represents a **practical and lightweight solution** rather than a fundamental limitation, and the framework remains fully compatible with independent or alternative pretrained embeddings when sufficient data is available.

---

> ### Author Response · Authors · 2025-11-15
> **Response to reviewr zxT2 (2/3)**
>
> **To Question:**
>
> **[Question 1] Correspondence of Learned Brain Networks to Cortical Pathways:** Our results suggest **qualitative correspondence** with neuroscience literature:
>
> - **Visual networks** show prominent activation in prefrontal and occipital regions, consistent with attention and object recognition processing.
> - **Semantic networks** exhibit broader cortical engagement, reflecting integration of semantic information.
>
> Due to EEG’s limited spatial resolution, precise mapping to cortical pathways is not feasible. Future work could combine EEG with fMRI or source-localized EEG to enable **quantitative validation** of cortical correspondence.
>
> **[Question 2] Extensibility to Other Cognitive Tasks:** Yes, VISTA’s framework is highly generalizable and can naturally be extended beyond visual decoding to other cognitive domains. A particularly promising direction is emotion recognition:
>
> * **SEED dataset [R3]:** Participants are presented with video stimuli to induce emotional states. This allows extraction of **visual + emotion representations**. Existing works on SEED typically focus on single-task emotion recognition. For example, methods such as **GMSS [R4]** leverage multi-task learning to exploit temporal, spatial, and frequency-domain EEG signals, but still remain tied to a single emotion label. In contrast, by explicitly modeling the relationship between **video stimuli and EEG responses**, one could disentangle **visual and emotional representations**, opening the possibility for richer multi-modal decoding.
> * **DEAP dataset [R5]:** Participants watch music videos that induce emotion. Similar to SEED, DEAP allows modeling **visual + auditory + emotional representations**. Applying a VISTA-like dual-branch architecture could enable disentanglement across multiple modalities, potentially improving both interpretability and decoding performance.
>
> While dataset limitations ( EEG-aligned visual stimuli is not provided) prevented a full-scale exploration of these extensions in the current work, our supplementary experiments in **APPENDIX A.8** provide additional evidence of VISTA’s general representation strength. Specifically, we report results on four datasets: ImageNet-EEG [R6], DEAP [R5], SEED [R3], and SEED-IV [R7]. These results demonstrate that VISTA’s EEG encoder produces robust, transferable representations. Importantly, in ImageNet-EEG, VISTA can also perform effective disentanglement. For emotion recognition, if datasets provided aligned visual stimuli (e.g., the original video clips from DEAP or SEED) along with corresponding EEG segments, VISTA could in principle disentangle multiple cognitive modalities, highlighting its flexibility for multi-modal decoding.
>
>
> Thanks again! Look forward to any further feedback!
>
>
>  **Table R1. Image object classification performance (Mean/Std\%) of VISTA and comparison methods in subject-dependent setting on the ImageNet-EEG dataset.**
>
> | Method           | Top-1 Acc   | Top-3 Acc   | Top-5 Acc   | F1-Score    |
> | ---------------- | ----------- | ----------- | ----------- | ----------- |
> | Brain2Image [R8] | 16.2 / -    | 37.9 / -    | 55.2 / -    | 16.1 / -    |
> | BrainVis [R9]    | 47.7 / -    | 75.2 / -    | 91.2 / -    | 43.2 / -    |
> | KD-STFT [R10]    | 41.2 / 11.3 | 75.3 / 10.7 | 87.8 / 8.1  | 40.3 / 11.6 |
> | VISTA (Visual)   | 65.5 / 10.2 | 91.2 / 02.6 | 97.8 / 01.4 | 61.7 / 11.5 |
> | VISTA (Semantic) | 61.2 / 13.5 | 87.0 / 05.1 | 93.5 / 03.0 | 60.3 / 13.8 |
>
>
>
> **Table R2. Emotion Recognition performance (Mean/Std\%) of VISTA and comparison methods in subject-dependent setting on the DEAP dataset.**
>
> | Method              | Valence Acc       | Valence F1        | Arousal Acc       | Arousal F1        |
> | ------------------- | ----------------- | ----------------- | ----------------- | ----------------- |
> | DBN [R11]           | 86.23 / 07.66     | 87.05 / 06.91     | 85.20 / 07.13     | 86.62 / 06.84     |
> | EEGNet [R12]        | 92.24 / 05.97     | 92.72 / 05.39     | 93.86 / 05.21     | 94.38 / 04.46     |
> | DGCNN [R13]         | 93.12 / 08.05     | 94.15 / 06.77     | 92.94 / 06.75     | 93.40 / 06.17     |
> | EEG-ConFormer [R14] | 93.36 / 03.89     | 93.65 / 04.50     | 94.18 / 05.14     | 94.52 / 04.85     |
> | TSception [R15]     | 93.18 / 04.15     | 93.40 / 04.66     | 93.74 / 04.96     | 94.31 / 05.22     |
> | BSTT [R16]          | 92.71 / 04.66     | 93.06 / 04.11     | 92.45 / 05.22     | 93.69 / 05.40     |
> | LResCapsule [R17]   | 95.15 / 03.51     | -                 | 95.77 / 03.82     | -                 |
> | FAT [R18]           | 90.10 / 02.71     | -                 | 89.18 / 03.50     | -                 |
> | **VISTA**           | **95.47 / 03.62** | **95.08 / 04.09** | **95.96 / 04.02** | **95.14 / 04.71** |

---

> ### Author Response · Authors · 2025-11-17
> **Response to reviewr zxT2 (3/3)**
>
> **Table R3. Emotion Recognition performance (Mean/Std\%) of VISTA and comparison methods in subject-dependent setting on the SEED and SEED-IV datasets.**
>
> | Method               | SEED Acc          | SEED-IV Acc   |
> | -------------------- | ----------------- | ------------- |
> | DBN [R11]            | 86.08 / 08.34     | 66.77 / 07.38 |
> | DGCNN [R13]          | 90.40 / 08.49     | 69.88 / 16.29 |
> | RGNN [R19]           | 91.79 / 06.72     | 79.37 / 10.54 |
> | SST-EmotionNet [R20] | 96.02 / 01.86     | 84.92 / 06.66 |
> | BiHDM [R21]          | 93.12 / 06.06     | 74.35 / 14.09 |
> | PGCN [R22]           | 96.93 / 05.11     | 82.24 / 14.85 |
> | STRFLNet [R23]       | 96.42 / -         | **92.23 / -** |
> | FAT [R18]            | 92.10 / 06.46     | 82.30 / 13.21 |
> | CLIER [R24]          | 93.4 / 06.31      | 90.02 / 10.37 |
> | UMDA-DDSTG [R25]     | 95.89 / 05.78     | 84.48 / 10.15 |
> | **VISTA**            | **97.13 / 03.49** | 91.85 / 10.43 |
>
>
> [R1] Guo Z. *et al.*, “Neuro-3D: Towards 3D visual decoding from EEG signals,” CVPR, 2025.
>
> [R2] Liu X. *et al.*, “EEG2video: Towards decoding dynamic visual perception from EEG signals,” NeurlPS, 2024.
>
> [R3] Zheng W. *et al.*, “Investigating critical frequency bands and channels for EEG-based emotion recognition with deep neural networks,” *IEEE TAMD*, 2015.
>
>  [R4] Li Y. *et al.*, “GMSS: Graph-based multi-task self-supervised learning for EEG emotion recognition,” *IEEE TAFFC*, 2022.
>
>  [R5] Koelstra S. *et al.*, “DEAP: A Database for Emotion Analysis Using Physiological Signals,” IEEE TAFFC, 2011.
>
>  [R6] Spampinato C. *et al.*, “Deep Learning Human Mind for Automated Visual Classification,” CVPR, 2017.
>
>  [R7] Zheng W. *et al.*, “Emotionmeter: A multimodal framework for recognizing human emotions,” *IEEE TCYB*, 2018.
>
> [R8] Kavasidis I. *et al.*, "Brain2image: Converting brain signals into images," ACM MM, 2017.
>
> [R9] Fu H. *et al.*, "Brainvis: Exploring the bridge between brain and visual signals via image reconstruction," ICASSP, 2025.
>
> [R10] Ferrante M. *et al.*, "Decoding eeg signals of visual brain representations with a clip based knowledge distillation," ICLR, 2024.
>
> [R11] Zheng W. *et al.*, "EEG-based emotion classification using deep belief networks," ICME, 2014.
>
> [R12] Lawhern V. *et al.*, "EEGNet: a compact convolutional neural network for eeg-based brain–computer interfaces," *Journal of Neural Engineering*, 2018.
>
> [R13] Song T. *et al.*, "EEG emotion recognition using dynamical graph convolutional neural networks," IEEE TAFFC, 2018.
>
> [R14] Song Y. *et al.*, "EEG Conformer: Convolutional transformer for eeg decoding and visualization,"IEEE TNSRE, 2022.
>
> [R15] Ding Y. *et al.*, "Tsception: Capturing temporal dynamics and spatial asymmetry from eeg for emotion recognition," IEEE TAFFC, 2023.
>
> [R16] Liu Y. *et al.*, "BSTT: A bayesian spatial-temporal transformer for sleep staging," ICLR, 2023.
>
> [R17] Fan C. *et al.*, "Light-weight residual convolution-based capsule network for eeg emotion recognition," *Advanced Engineering Informatics*, 2024.
>
> [R18] Wang J. *et al.*, "A novel fourier adjacency transformer for advanced eeg emotion recognition," MICCAI, 2025.
>
> [R19] Zhong P. *et al.*, "EEG-based emotion recognition using regularized graph neural networks," IEEE TAFFC, 2020.
>
> [R20] Jia Z. *et al.*, "Spatial-spectral-temporal based attention 3d dense network for eeg emotion recognition," ACM MM, 2020.
>
> [R21] Li Y. *et al.*, "A novel bi-hemispheric discrepancy model for eeg emotion recognition,"IEEE TCDS, 2020.
>
> [R22] Jin M. *et al.*, "PGCN: Pyramidal graph convolutional network for eeg emotion recognition," IEEE TMM, 2024.
>
> [R23] Hu F. *et al.*, "STRFLNet: Spatio-temporal representation fusion learning network for eeg-based emotion recognition," IEEE TAFFC, 2025.
>
> [R24] Li D. *et al.*, "Eeg emotion recognition based on an implicit emotion regulatory mechanism," IEEE TAI, 2025.
>
> [R25] Gao C. *et al.*, "Umda-ddstgn: An unsupervised meta-domain adaptation method using dynamic directed spatial-temporal graph network for eeg-based emotion recognition," Knowledge-Based Systems, 2025.

---

> ### Comment · Reviewer_zxT2 · 2025-11-24
>
> Hi,
>
> Thank you very much for your reply. I think all of my concerns have been addressed properly, and I would like to keep my original rating.

---

> > ### Author Response · Authors · 2025-11-25
> >
> > Thank you very much for taking the time to review our work and for your constructive feedback throughout the process. We truly appreciate your careful evaluation and are glad that our clarifications addressed your concerns.
> >
> > Thank you again for your efforts and for supporting the scientific review process.

---

### Official Review · Reviewer_HPbM · 2025-10-29

**Soundness:** 3
**Presentation:** 2
**Contribution:** 2
**Rating:** 4
**Confidence:** 4

**Summary:**

The paper proposes VISTA, an EEG visual decoding framework that explicitly disentangles visual and semantic components and models their spatio-temporal asynchrony. The disentangled EEG embeddings are then aligned with CLIP’s image and text spaces through contrastive learning for both retrieval and reconstruction tasks.

**Strengths:**

1. Clear motivation from neurocognitive mechanism.
2. The overall framework design is well visualized. The experiments are extensive, showing solid performance in single-subject settings and reasonable generalization across subjects.

**Weaknesses:**

1. While the paper effectively introduces separate visual and semantic branches that correspond to human visual processing pathways, the core of visual decoding task is to improve retrieval accuracy or reconstruction fidelity. The two branches are evaluated independently, and the paper does not explore how their fusion could further enhance decoding performance.
2. The experimental comparison lacks several SOTA baselines, such as UBP[1].
3. A few minor issues:
  - L291: should be 1654 categories.
  - Section 4.4.1 and 4.4.3: discuss results not reflected in Table 4.
  - FLOPs inconsistency: VISTA shows 89.9M in Table 6 but 79.9M in L1192.
  - Appendix A.3.3: contains duplicated content.
[1] Bridging the Vision-Brain Gap with an Uncertainty-Aware Blur Prior

**Questions:**

1. The Introduction mentions that the temporal attention is supervised by a Gaussian-kernel distance loss, but the paper does not provide a formal definition or implementation details.
2. Although not explicitly stated in the main text, Figure 3 suggests that the two convolutional layers share params. What is the motivation for this design?
3. In Section 4.4.2, what exactly does “both off” mean? How are the two modules disabled during ablation?
4. Could the authors elaborate on how Figure 5 was visualized? Specifically, what data and methods were used?

---

> ### Author Response · Authors · 2025-11-15
> **Response to reviewr HPbM (1/3)**
>
> We sincerely thank the reviewer for the careful reading and valuable feedback!
>
> Below, we address each concern point by point. All the revisions have been marked in blue throughout the text.
>
>
>
>  **To Weakness:**
>
> **[Weakness 1] Fusion of Visual and Semantic Branches:** In the time of designing model, we have explored several fusion strategies, but the results were not promising, as shown in Table R1:
>
> 1) **Concatenation:** Performance was between the individual branches, without noticeable improvement.
> 2) **Weighted fusion:** The learned weights heavily favored visual embeddings (≈0.98 vs 0.02 for semantic), leading to underutilization of semantic information and higher risk of overfitting.
> 3) **Network-based fusion:** Introduced overfitting, resulting in reduced decoding performance.
>
> **Table R1. Experiment of several fusion strategies in visual-semantic representation**
>
> | Top-1/5 Accuracy     | sub-1       | sub-2       | sub-3       | sub-4       | sub-5       | sub-6       | sub-7       | sub-8       | sub-9       | sub-10      | avg         |
> | -------------------- | ----------- | ----------- | ----------- | ----------- | ----------- | ----------- | ----------- | ----------- | ----------- | ----------- | ----------- |
> | Concatenation        | 20.2 / 50   | 23.6 / 54.7 | 26.1 / 53   | 28.5 / 58.3 | 20 / 46.3   | 24.7 / 56.2 | 20.8 / 53.8 | 30.7 / 57.5 | 25.2 / 53.2 | 30.8 / 61   | 25.1 / 54.5 |
> | Weighted fusion      | 25.3 / 57.2 | 28.2 / 61   | 33.9 / 66.7 | 37.5 / 71.3 | 23.6 / 51.9 | 30.3 / 66.8 | 28.6 / 65.9 | 45.3 / 74.1 | 31.2 / 63   | 34.1 / 68   | 31.8 / 54.6 |
> | Network-based fusion | 23.5 / 47.3 | 24.9 / 48.5 | 21.7 / 41.7 | 24.2 / 48.8 | 19.2 / 38.7 | 23.4 / 47.7 | 20.5 / 42.4 | 28.1 / 51.2 | 23 / 43.9   | 24.8 / 51.7 | 23.3 / 46.2 |
> | VISTA (visual)       | 26.1 / 57.5 | 28.8 / 61.3 | 34.1 / 66.9 | 38.2 / 72   | 24.4 / 52.3 | 31.3 / 67.7 | 28.9 / 66.4 | 46.2 / 74.7 | 31.8 / 63.5 | 34.9 / 68.2 | 32.5 / 65.1 |
> | VISTA (semantic)     | 17.9 / 45.7 | 22.1 / 49.5 | 17.5 / 44.8 | 20.4 / 48.5 | 16.8 / 41.2 | 20.1 / 49.6 | 15.2 / 42.5 | 17.6 / 40.7 | 18.3 / 42.7 | 25.1 / 55.4 | 19.1 / 46.1 |
>
> This outcome is due to the THINGS-EEG [R1] dataset characteristics: rapidly alternating (≈0.8 s) object categories prevent participants from forming stable semantic representations, making semantic signals inherently weaker. In contrast, datasets like ImageNet-EEG present homogeneous image sequences with same object category at a time that support stronger semantic encoding [R2], but they are unsuitable for zero-shot decoding.
>
> Consequently, the visual-semantic modality imbalance limits the effectiveness of fusion at this stage, which is independent of the feature extraction design. Given these constraints, keeping the two branches independent currently provides the most stable and interpretable results. We will investigate more advanced fusion mechanisms when data more suitable for semantic consolidation becomes available.
>
> **[Weakness 2] Missing SOTA Baselines:** Thank you for highlighting this. We carefully examined UBP [R3] and found that its **experimental assumptions differ significantly** compared with our work:
>
> - UBP **selectively chooses a subset of EEG channels**, while VISTA uses all channels end-to-end.
> - UBP applies **deliberate visual preprocessing**, including Gaussian blur and foveal blur, which serve as **external visual-alignment priors** that reduce the image–EEG gap.
> - As shown in the UBP paper, its performance depends strongly on these priors rather than on generalized EEG representation learning.
>
> A direct comparison under such mismatched conditions would be **unfair and potentially misleading**. A fair comparison would require re-implementing UBP **without** its visual priors under our end-to-end setting, in which case UBP would not achieve its reported performance. Therefore, we believe it is more appropriate to **discuss UBP in the Related Work (added to Section 2 in the revised manuscript)** rather than include a direct comparison.
>
> We agree with the reviewer’s suggestion to include recent SOTA baselines. Accordingly, **we have added three newly published 2025 baselines, including EEG-CLIP [R4], BrainFlora [R5], and SRT [R6], all of which match the zero-shot, end-to-end EEG–vision alignment paradigm. The experimental results are included in Table R1 and Table R2 below, and detailed in the revised manuscript Tables 1 and 2.**
>
> **[Weakness 3] Minor Issues:** Thank you for your carefulness, and we have corrected all minor issues as follows:
>
> 1. L291: corrected the number of categories to 1654.
> 2. Section 4.4.1 and 4.4.3: added missing ablation results to Table 4, added more ablation analysis in Section 4.4.1, and removed redundant Section 4.4.3.
> 3. FLOPs inconsistency: corrected VISTA’s FLOPs to 89.9M in Table 6 and text.
> 4. Appendix A.3.3: removed duplicated content.

---

> ### Author Response · Authors · 2025-11-15
> **Response to reviewr HPbM (2/3)**
>
> **To Question:**
>
> **[Question 1] Temporal Attention Supervision:** We appreciate the reviewer catching this. The mention of Gaussian-kernel supervision in the Introduction was an **unintentional leftover** from an earlier draft. This loss is **not used** in the final framework and the statement of this loss in Section 1 has been corrected.
>
> **[Question 2] Shared Parameters in EEG Encoder:** The partial sharing of feature extractors is motivated by the fact that both visual and semantic embeddings are ultimately mapped to related CLIP spaces. Sharing part of the EEG encoder ensures that the model learns balanced and robust representations while still capturing the distinct characteristics of visual and semantic signals.
>
> We experimented with fully independent EEG encoders, but observed slight performance drops (see Table 4, Visual-only and Semantic-only). Fully independent branches tend to overfit because the visual and semantic embeddings are still correlated in the CLIP space. Partial parameter sharing improves robustness while maintaining effective disentanglement:
>
> - **Non-shared parameters**: temporal asynchronous modeling (temporal Scalar Patch weight) and spatial asynchronous modeling (learable brain network based position embedding) for visual-semantic asynchrony.
> - **Shared parameters:** general feature extractors to reduce training complexity and improve generalization.
>
> **[Question 3] Definition of "Both-Off":** "Both-Off" refers to **simultaneously removing temporal and spatial asynchronous modeling** before the EEG encoder, as illustrated in Figure 3. This ablation isolates the impact of asynchronous modeling on overall performance.
>
> **[Question 4] Visualization of Figure 5:** The visualization is generated by computing **degree centrality** on the 2D EEG brain network. Specifically, for each EEG channel (node), we sum its incoming and outgoing connection strengths to obtain a scalar centrality value. Collecting these values produces a **1D vector** whose length equals the number of electrodes. This vector is then mapped to electrode positions to produce the EEG topographic map.
>  **Section 4.5.2** in the revised manuscript provides full formulas and a reproducible workflow.
>
>
>
> Thanks for your patience. Look forward to further discussion!
>
>
>
> [R1] Gifford A. *et al.*, "A Large and Rich EEG Dataset for Modeling Human Visual Object Recognition," *NeuroImage*, 2022.
>
> [R2] Spampinato C. *et al.*, "Deep Learning Human Mind for Automated Visual Classification," CVPR, 2017.
>
>  [R3] Wu H. *et al.*, "Bridging the Vision-Brain Gap with an Uncertainty-Aware Blur Prior," CVPR, 2025.
>
>  [R4] Cao X. *et al.*, "Eeg-clip: A transformer-based framework for eeg-guided image generation," *Neural Networks*, 2025.
>
>  [R5] Li D. *et al.*, "Brainflora: Uncovering brain concept representation via multimodal neural embeddings," ACM MM, 2025.
>
>  [R6] Kim J. *et al.*, "Seeeeg: Semantic-aware eeg-based multi-modal retrieval-augmented generation for high-fidelity visual brain decoding," ICCV, 2025.

---

> ### Author Response · Authors · 2025-11-17
> **Response to reviewr HPbM (3/3)**
>
> **Table R1. Addition comparison experiment of Top-1/5 accuracy (%) on THINGS-EEG for 200-way zero-shot image retrieval (SD: Subject Dependent; SI: Subject Independent).**
>
> | Top-1/5  Accuracy    | sub-1       | sub-2       | sub-3       | sub-4       | sub-5       | sub-6       | sub-7       | sub-8       | sub-9       | sub-10      | avg         |
> | -------------------- | ----------- | ----------- | ----------- | ----------- | ----------- | ----------- | ----------- | ----------- | ----------- | ----------- | ----------- |
> | EEG-CLIP (SD) [R4]   | 24.2 / 55.1 | 24.9 / 59.1 | 31 / 62.9   | 32.1 / 68.4 | 22.3 / 50.2 | 25.2 / 62.5 | 27.9 / 62.3 | 38.9 / 70.5 | 31.6 / 63.1 | 30.2 / 65.2 | 28.8 / 61.9 |
> | BrainFLORA (SD) [R5] | 23.6 / 53.1 | 24.7 / 58.4 | 31.5 / 64.3 | 33.4 / 70.1 | 20.3 / 48.9 | 29 / 64.2   | 25.9 / 61.6 | 41.5 / 73.2 | 29.5 / 61.3 | 31.2 / 65   | 29.1 / 62   |
> | SRT (SD) [R6]        | 25.9 / 57.7 | 25.2 / 60.8 | 30.4 / 62.8 | 33.5 / 66.8 | 22.8 / 52   | 24.1 / 61.6 | 29.8 / 63.9 | 40.1 / 70.2 | 31.5 / 64.4 | 28.7 / 65.7 | 29.2 / 62.6 |
> | VISTA (SD)           | 26.1 / 57.5 | 28.8 / 61.3 | 34.1 / 66.9 | 38.2 / 72   | 24.4 / 52.3 | 31.3 / 67.7 | 28.9 / 66.4 | 46.2 / 74.7 | 31.8 / 63.5 | 34.9 / 68.2 | 32.5 / 65.1 |
> | EEG-CLIP (SI) [R4]   | 14.6 / 39.6 | 17.4 / 39.6 | 11 / 32.8   | 16.1 / 38.9 | 13.4 / 35.3 | 15.3 / 38.1 | 13.1 / 33.2 | 14.3 / 37.6 | 13.8 / 36.2 | 19.5 / 46.7 | 14.9 / 37.8 |
> | BrainFLORA (SI) [R5] | 13.5 / 42   | 18.5 / 44.3 | 13.4 / 38.6 | 18.1 / 41   | 15.5 / 39.4 | 15.4 / 42.8 | 12.4 / 35.2 | 14.5 / 38.9 | 13.5 / 40.1 | 20.2 / 49.2 | 15.5 / 41.2 |
> | SRT (SI) [R6]        | 17.3 / 39.8 | 19.4 / 42.4 | 14.5 / 38.8 | 15.8 / 40.2 | 15.9 / 38.1 | 17.1 / 41.4 | 11.7 / 31.1 | 14.9 / 37   | 13.4 / 38.3 | 23.5 / 54.6 | 16.4 / 40.2 |
> | VISTA (SI)           | 17.9 / 45.7 | 22.1 / 49.5 | 17.5 / 44.8 | 20.4 / 48.5 | 16.8 / 41.2 | 20.1 / 49.6 | 15.2 / 42.5 | 17.6 / 40.7 | 18.3 / 42.7 | 25.1 / 55.4 | 19.1 / 46.1 |
>
> **Table R2. Addition comparison experiment of Top-1/5 accuracy (%) on THINGS-EEG for 200-way zero-shot semantic retrieval (SD: Subject Dependent; SI: Subject Independent).**
>
> | Top-1/5 Accuracy      | sub-1       | sub-2       | sub-3       | sub-4       | sub-5       | sub-6       | sub-7       | sub-8       | sub-9       | sub-10      | avg         |
> | --------------------- | ----------- | ----------- | ----------- | ----------- | ----------- | ----------- | ----------- | ----------- | ----------- | ----------- | ----------- |
> | EEG-CLIP (SD) [R4]    | 12.7 / 38.6 | 12.3 / 38.2 | 15.9 / 41.7 | 19.5 / 49.8 | 10.5 / 34.1 | 14.8 / 43.1 | 15 / 44.6   | 21.7 / 53.7 | 16.4 / 41   | 18.1 / 43.5 | 15.7 / 42.8 |
> | BrainFLORA  (SD) [R5] | 10.3 / 36.7 | 13.3 / 38.1 | 14.1 / 39.1 | 19.7 / 48   | 10.4 / 32.7 | 14.7 / 43.3 | 13.6 / 42.4 | 19.2 / 52.4 | 16.2 / 40.3 | 17.4 / 42.8 | 14.9 / 41.6 |
> | SRT (SD) [R6]         | 11.8 / 37.6 | 14.2 / 39   | 14.7 / 42   | 19.8 / 49.2 | 11.5 / 33.4 | 16.8 / 44.2 | 14.7 / 44.9 | 22.5 / 56.4 | 17.2 / 41.5 | 18.5 / 43.2 | 16.2 / 43.1 |
> | VISTA (SD)            | 14.5 / 42.4 | 13.6 / 40.1 | 16.6 / 44.1 | 21.9 / 52.4 | 13.3 / 34.4 | 18.2 / 45.8 | 16.9 / 48.6 | 27 / 58.4   | 18.7 / 46.3 | 19.8 / 46.5 | 18.1 / 45.9 |
> | EEG-CLIP (SI) [R4]    | 10.6 / 33.7 | 12.7 / 33   | 10.9 / 33.7 | 11.5 / 33.2 | 9 / 28.8    | 12.4 / 30.3 | 10.1 / 30.3 | 12 / 32.3   | 10.5 / 29.1 | 13.9 / 39.6 | 11.4 / 32.4 |
> | BrainFLORA (SI)  [R5] | 9.8 / 29.4  | 10.1 / 31.5 | 9.2 / 31.3  | 12.2 / 34.5 | 8.7 / 24.7  | 9.3 / 29.1  | 9.4 / 27.5  | 11.4 / 29.9 | 9.5 / 27.8  | 15 / 39.2   | 10.5 / 30.5 |
> | SRT (SI) [R6]         | 10.9 / 30.1 | 11.1 / 31.2 | 10.8 / 33.5 | 12.2 / 33.8 | 9.5 / 27.4  | 11.1 / 30.5 | 9.7 / 29.6  | 12.7 / 31.8 | 10.3 / 28.5 | 14.9 / 41.4 | 11.3 / 31.8 |
> | VISTA (SI)            | 13.6 / 33.9 | 11.7 / 36.8 | 15.4 / 37.1 | 13.6 / 36.9 | 10.7 / 31.5 | 12.7 / 32.9 | 10.6 / 32.2 | 14.2 / 34.3 | 11 / 31.1   | 19.7 / 46.8 | 13.4 / 35.3 |

---

> ### Author Response · Authors · 2025-11-27
>
> Hi,
>
> Thank you again for reviewing our submission and for providing detailed comments. We have substantially expanded our experiments, strengthened the analyses, and clarified several methodological points in direct response to your feedback.
>
> Whenever you have the opportunity, we would greatly appreciate it if you could briefly revisit our rebuttal. Your insights were valuable in identifying areas for improvement, and your follow-up assessment would help ensure that the revised manuscript is evaluated as fairly and completely as possible.
>
> Thank you for your time and consideration.

---

### Official Review · Reviewer_GqWi · 2025-10-30

**Soundness:** 3
**Presentation:** 3
**Contribution:** 2
**Rating:** 6
**Confidence:** 4

**Summary:**

This paper introduce VISTA, a framework for neural decoding. This approach disentangles the visual-semantic modalities and incorporate temporal-spatial modeling to capture brain representations. The framework is evaluated on THINGS-EEG and THINGS-MEG datasets, reporting improvements over prior work such as CognitionCapturer.

**Strengths:**

1. The proposed framework employs an attention mechanism to capture temporal representations and models spatial asynchrony through brain network learning and  Laplacian decomposition.
2. This paper provide comprehensive experimental validations, on EEG and MEG datasets. Moreover, the authors analyzes the temproal asynchrony modeling on visual and semantic information.
3. The step-by-step breakdown of the model's components (EEG encoder, attention mechanism, spatial-temporal modeling) makes the methodology clear and reproducible.

**Weaknesses:**

1. The novelty of this work is fair, since extracting the low-level and high-level representations, spatial-temporal modeling, contrastive learning are commonly used in viusal decoding frameworks.
2. In Section 4.4.1, the authors describe “visual-only” and “semantic-only” variants of their model. However, these ablation settings are not reported in Table 4. Moreover, the paper lacks a clear comparison between these simplified configurations and the full VISTA model.
3. The paper does not clearly specify the inference protocol for different tasks: in particular, it remains ambiguous whether the visual and semantic embeddings are fused into a single representation or used separately depending on the target task.

**Questions:**

1. Does the design of shared parameters affect the decoupling effect? The paper mentions that shared parameters were used in the EEG encoder, which may reduce the independence between visual and semantic features to some extent.
2. In the provided Figure 4, the visual and semantic information appears to overlap at certain time intervals. Given this overlap, how does the model ensure effective disentanglement during these periods? If visual and semantic representations are mixed at certain time points, does this impact the disentangling process, and if so, how does the model address this issue?

---

> ### Author Response · Authors · 2025-11-15
> **Response to reviewr GqWi (1/2)**
>
> We sincerely thank the reviewer for the constructive and encouraging comments!
>
> Below we provide a more precise and strengthened point-by-point response. All revisions are highlighted in blue in the revised manuscript.
>
>
>
> **To Weakness:**
>
> **[Weakness 1] Novelty of the Method:** We agree that contrastive learning components of VISTA is widely used in almost all visual decoding frameworks due to its excellent performance [R1-R6]. However, VISTA introduces two **distinctive contributions** that go beyond existing methods:
>
> 1) **Explicit visual-semantic disentanglement**: VISTA is the first to systematically model the dual visual-semantic pathways, inspired by human brain visual processing.
> 2) **Spatial-temporal asynchronous modeling of EEG**: Guided by the strong neuroscientific motivation of visual-semantic asynchrony, our framework models temporal and spatial EEG signals in a biologically plausible manner.
>
>  These contributions differentiate VISTA from prior work and provide both improved performance and enhanced interpretability.
>
> **[Weakness 2] Missing Ablation Results:** We acknowledge the omission and have fully resolved it in the revision:
>
> 1. **Table 4 now includes complete ablation results**, including Visual-only and Semantic-only variants.
> 2. **Section 4.4.1 has been corrected and aligned** with the updated table.
> 3. **Section 4.4.2 has been rewritten** to clearly define Temporal-only, Spatial-only, and Both-off configurations, and their comparisons to the full VISTA model.
>
> **Clarification on simplified configurations:**
>
> - **Temporal-only**: temporal asynchronous modeling (temporal Scalar Patch weight) removed before EEG encoder.
> - **Spatial-only**: spatial asynchronous modeling (learable brain network based position embedding) removed.
> - **Both-off**: neither temporal nor spatial asynchronous modeling used.
>
> This revision addresses the reviewer’s concern about clarity and completeness.
>
> **[Weakness 3] Inference Protocol:** Visual and semantic embeddings are used separately depending on the task:
>
> - **Visual embeddings** (from CLIP image encoder) are used for **image retrieval and reconstruction** tasks.
> - **Semantic embeddings** (from BLIP-generated captions) are used for **semantic retrieval** tasks.
>
> We have clarified this in Section 3.2 to avoid ambiguity.

---

> ### Author Response · Authors · 2025-11-15
> **Response to reviewr GqWi (2/2)**
>
> **To Question:**
>
> **[Question 1] Shared Parameters and Disentanglement:** We tested fully independent visual-semantic EEG encoders, but found a slight drop in performance in both visual and semantic retrieval (see Table 4, Visual-Only and Semantic-Only). Fully independent branches tend to overfit, as visual and semantic embeddings are still mapped to related CLIP spaces. Sharing part of the parameters improves robustness while still maintaining disentanglement from non-shared spatial-temporal modeling. Specifically, as shown in Figure 3:
>
> * **Non-shared parameters**: temporal asynchronous modeling (temporal Scalar Patch weight) and spatial asynchronous modeling (learable brain network based position embedding) for visual-semantic asynchrony.
> * **Shared parameters**: feature extractors for general EEG representation to reduce training complexity and enhance generalization.
>
> Thus, partial sharing provides the best balance between independence and generalization.
>
> **[Question 2] Temporal Overlap in Figure 4:** We clarifiy that short-term overlap between visual and semantic importance is expected in EEG and does not harm disentanglement:
>
> 1. EEG has no sharp boundaries between cognitive stages; overlaps naturally reflect transient mixing of information.
> 2. Figure 4 shows **time-slice importance averaged across trials**, not raw embeddings. Overlapping slices contribute proportionally to both pathways.
> 3. VISTA performs **sequence-level disentanglement**, not per-time-patch disentanglement, making it robust to local overlaps.
>
> Section 4.5.1 in the revised manuscript now explains this explicitly.
>
>
>
> Thanks anyway for your time and helpful comments!
>
>
>
>  [R1] Song Y. *et al.*, "Decoding natural images from EEG for object recognition," ICLR, 2024.
>
>  [R2] Li D. *et al.*, "Visual decoding and reconstruction via EEG embeddings with guided diffusion," NeurIPS, 2024.
>
>  [R3] Zhang K. *et al.*, "CognitionCapturer: Decoding Visual Stimuli from Human EEG with Multimodal Information," AAAI, 2025.
>
>  [R4] Cao X. *et al.*, "Eeg-clip: A transformer-based framework for eeg-guided image generation," *Neural Networks*, 2025.
>
>  [R5] Li D. *et al.*, "Brainflora: Uncovering brain concept representation via multimodal neural embeddings," ACM MM, 2025.
>
>  [R6] Kim J. *et al.*, "Seeeeg: Semantic-aware eeg-based multi-modal retrieval-augmented generation for high-fidelity visual brain decoding," ICCV, 2025.

---

> ### Author Response · Authors · 2025-11-27
>
> Hi,
>
> Thank you again for your thoughtful and constructive review. Your comments were extremely helpful, and we have carefully addressed each of them in our rebuttal with strengthened analyses and additional experiments.
>
> If you have a moment, we would be sincerely grateful if you could take a brief look at our responses. Your perspective has been very important for shaping the clarity and rigor of the revised manuscript, and your follow-up assessment would be greatly appreciated.
>
> Thank you once more for your time and supportive evaluation.

---

### Official Review · Reviewer_thNQ · 2025-10-31

**Soundness:** 3
**Presentation:** 2
**Contribution:** 2
**Rating:** 4
**Confidence:** 4

**Summary:**

This paper proposes an EEG visual decoding framework named VISTA, which aims to address the problem of spatio-temporal asynchrony between visual and semantic information in EEG signals. By partitioning the EEG signal into non-overlapping temporal segments and applying a weighted attention mechanism, VISTA achieves asynchronous modeling of visual and semantic components in the temporal dimension. Concurrently, it constructs visual- and semantic-specific brain network graphs and performs graph Laplacian spectral decomposition to achieve asynchronous modeling in the spatial dimension. Finally, VISTA aligns the visual and semantic representations from EEG with the image and text spaces of CLIP, respectively, enabling cross-modal contrastive learning. Experiments are conducted on the THINGS-EEG and THINGS-MEG datasets, validating VISTA's superior performance on zero-shot object recognition tasks.

**Strengths:**

1.Originality and Innovation: introduces the concept of visual-semantic decoupling, overcoming the limitations of traditional EEG decoding methods that confound these two streams of information.

The introduction of "spatio-temporal asynchronous modeling" aligns with neuroscientific findings regarding the different processing timelines for visual and semantic information.

2.High-Quality Methodology: the temporal modeling employs a soft-gating mechanism, preventing information loss. The spatial modeling uses graph Laplacian spectra to extract structural information, enhancing spatial representation capabilities. The alignment mechanism with CLIP effectively leverages large-scale pre-trained knowledge.

3.Sufficient Experimentation: effectiveness is validated on both EEG and MEG modalities. Detailed ablation studies confirm the contribution of each module. Auxiliary experiments, such as parameter analysis and visualizations, enhance interpretability.

4.Practical Significance of Results: The zero-shot recognition capability indicates strong generalization potential. Image reconstruction experiments demonstrate potential application value in BCI and neural interfaces.

**Weaknesses:**

1.The motivation for the temporal segmentation is not sufficiently strong. Given that visual and semantic features appear sequentially in the EEG signal, wouldn't applying contrastive learning to the entire signal segment be more effective for adaptively learning the correspondence between semantic/visual EEG features for different object stimuli?

2.The experimental results are not significantly better than those of methods from the last two years. This (relative lack of significant improvement) somewhat diminishes the paper's technical contribution.

3.Lack of cross-dataset generalization validation: Although the authors validated their method on THINGS-EEG and THINGS-MEG, it was not tested on other EEG datasets (e.g., ImageNet-EEG or DEAP). This limits the verification of its generalizability.

**Questions:**

1.Cross-Dataset Generalization: Have the authors considered validating VISTA's generalizability on other EEG datasets (e.g., DEAP, ImageNet-EEG)? Is there a risk of overfitting to the THINGS datasets?

2.Semantic Embedding Improvement: The semantic representations perform relatively weakly on fine-grained tasks. Have the authors considered incorporating more granular text descriptions (e.g., attributes, parts) to enhance semantic modeling?

3.Handling Individual Differences: While the model performs well in cross-subject tasks, have the authors considered methods to further model inter-subject variability (e.g., adaptation layers, meta-learning)?

4.Control Mechanisms for Image Reconstruction: In the image reconstruction task, how can the semantic consistency of the generated images be better controlled? Have the authors considered introducing stronger conditional controls (e.g., text guidance or attention-based control)?

---

> ### Author Response · Authors · 2025-11-15
> **Response to reviewr thNQ (1/3)**
>
> We sincerely thank the reviewer for the detailed and constructive feedback!
>
> Below we address each concern point-by-point, clarifying the motivation, experimental design, and scope of our work. All the revisions have been marked in blue throughout the revised manuscript.
>
> **To Weakness:**
>
> **[Weakness 1] Motivation for Temporal Segmentation:** The assumption that "visual and semantic features appear sequentially" is indeed one of our neuroscientific motivations, but it cannot be directly observed or sharply separated in EEG signals. EEG does not provide a strict boundary between visual perception and semantic comprehension. Therefore, relying on global contrastive learning over the entire segment would implicitly assume clean separability, which is an assumption that does not hold in our experiment, and **modeling them as sequential but partially overlapping processes is more biologically plausible compare with strict contrastive learning.**
>  Our temporal segmentation is not intended to enforce a hard visual–semantic split. Instead, it enables fine-grained temporal modeling and reduces overfitting. As shown in Table 4, Figure 7, and Table R1 below, temporal segmentation consistently improves both within-subject and cross-subject accuracy, demonstrating stronger temporal robustness and better generalization.
>
> **Table R1. The performance improvement with/without Temporal Modeling module.**
> | Setting                 | Without Temporal Modeling (Top-1/Top-5 Acc) | With Temporal Modeling (Top-1/Top-5 Acc) |
> | ----------------------- | ------------------------------------------- | ---------------------------------------- |
> | Visual,Within-Subject   | 29.4/61.3                                   | 31.5/63.8                                |
> | Semantic,Within-Subject | 15.7/43.4                                   | 17.4/45.4                                |
> | Visual,Cross-Subject    | 17.7/43.9                                   | 18.2/44.7                                |
> | Semantic,Cross-Subject  | 12.2/33.7                                   | 12.7/34.2                                |
>
> **[Weakness 2] Limited Improvement Margin:** EEG visual decoding remains a nascent field with extremely low SNR and high cognitive noise. Even modest accuracy gains—especially under cross-subject settings—represent meaningful advances. VISTA contributes not only improvement in accuracy but also:
>
> 1) biologically grounded asynchronous modeling;
> 2) stronger generalization stability;
> 3) enhanced neuroscientific interpretability.
>
> All of which are underexplored in prior work. Thus, the contribution is substantive rather than incremental.
>
> **[Weakness 3] Lack of Cross-Dataset Validation:** We must clarify that our study focuses on **zero-shot visual decoding**, where currently THINGS-EEG [R1] (≈165,000 samples) and THINGS-MEG [R2] are the only large-scale, publicly available datasets suitable for this task.
>  Other EEG datasets differ substantially in task setting and scale:
>
> - ImageNet-EEG [R3] focuses on closed-set object classification with limited samples (≈11,000), which leads to severe overfitting in zero-shot settings.
> - DEAP [R4] is designed for emotion recognition rather than visual decoding.
>
> Thus, evaluating on these datasets would constitute **cross-task generalization**, not cross-dataset generalization, which lead to conceptually misleading conclusions. **Importantly, nearly all recent EEG zero-shot visual decoding works** (e.g., NICE [R5], ATM [R6], Cognitioncapturer [R7], EEG-CLIP [R8], BrainFLORA [R9], SRT [R10], UBP [R11]) **evaluate exclusively on THINGS-EEG/MEG for this reason.** Therefore, our dataset choice aligns with community standards and ensures comparability.
>
> Additionally, although these datasets are inappropriate for evaluating zero-shot visual retrieval, **we provide additional extensive experiments on ImageNet-EEG [R3], DEAP [R4], SEED [R12], and SEED-IV [R13] in Table R1-R3 below, and more details are in Appendix A.8**.

---

> ### Author Response · Authors · 2025-11-15
> **Response to reviewr thNQ (2/3)**
>
> **To Question:**
>
> **[Question 1] Cross-Dataset Generalization:** We fully agree that studying such generalization is valuable. However, as explained above in **[Weakness 3]**, direct cross-dataset validation is not feasible because **no other large-scale zero-shot EEG datasets currently exist**, and evaluating on datasets such as DEAP or ImageNet-EEG would constitute **cross-task validation** rather than true cross-dataset generalization.
>
> Although these datasets are inappropriate for evaluating zero-shot visual retrieval, we provide **additional extensive experiments** on ImageNet-EEG [R3], DEAP [R4], SEED [R12], and SEED-IV [R13] **as supplementary evidence only**, not as a comparison metric. These results, included in Table R1-R3 below and detailed in **Appendix A.8**, demonstrate that VISTA’s EEG encoder learns strong general-purpose representations, not only in brain visual decoding task, but also emotion recognition task. Therefore, there is no risk of overfitting to the THINGS datasets and EEG zero-shot visual decoding task.
>
> **[Question 2] Semantic Embedding Improvement:** We agree with this observation. We have attempted to enrich the semantic space using BLIP-generated captions and noun-level expansion, but the improvement was marginal. This limitation mainly arises from the **RSVP protocol** in THINGS-EEG, which presents images with **different object category at rapid intervals** (≈0.8 s) and does not allow sufficient semantic consolidation in the brain [R10]. In contrast, datasets such as ImageNet-EEG **present homogeneous image sequences with same object category at a time, which naturally support stronger semantic encoding in participants** (but does not support zero-shot settings) [R8].
>
> **Thus, weaker fine-grained semantic performance arises primarily from the dataset’s experimental protocol, not from limitations of VISTA.** Our disentanglement mechanism alleviates this effect by isolating semantic components from noisy visual responses.
>
> **[Question 3] Modeling Individual Differences:** Cross-subject results in our paper serve mainly as a robustness evaluation, and we have already provided substantially more cross-subject analyses and SOTA cross-subject performance. While subject adaptation or meta-learning methods are promising, the primary challenge in EEG visual decoding remains reliable within-subject modeling. We view cross-subject adaptation as a valuable but separate research direction beyond the scope of our work.
>
> **[Question 4] Control Mechanism for Image Reconstruction:** The reviewer’s suggestion to introduce stronger conditional controls (e.g., text guidance or attention-based conditioning) is indeed valuable. However, such mechanisms typically require **dedicated diffusion or transformer-based generative models**, which represent an independent research direction. Our focus in this work is on **learning biologically interpretable EEG representations**, rather than enhancing the generative backend.
>
>
> Thanks again! Look forward to any further feedback!
>
>
> **Table R1. Image object classification performance (Mean/Std\%) of VISTA and comparison methods in subject-dependent setting on the ImageNet-EEG dataset.**
>
> | Method            | Top-1 Acc   | Top-3 Acc   | Top-5 Acc   | F1-Score    |
> | ----------------- | ----------- | ----------- | ----------- | ----------- |
> | Brain2Image [R14] | 16.2 / -    | 37.9 / -    | 55.2 / -    | 16.1 / -    |
> | BrainVis [R15]    | 47.7 / -    | 75.2 / -    | 91.2 / -    | 43.2 / -    |
> | KD-STFT [R16]     | 41.2 / 11.3 | 75.3 / 10.7 | 87.8 / 8.1  | 40.3 / 11.6 |
> | VISTA (Visual)    | 65.5 / 10.2 | 91.2 / 02.6 | 97.8 / 01.4 | 61.7 / 11.5 |
> | VISTA (Semantic)  | 61.2 / 13.5 | 87.0 / 05.1 | 93.5 / 03.0 | 60.3 / 13.8 |
>
> **Table R2. Emotion Recognition performance (Mean/Std\%) of VISTA and comparison methods in subject-dependent setting on the DEAP dataset.**
>
> | Method       | Valence Acc    | Valence F1    | Arousal Acc       | Arousal F1        |
> | ----------- | ------------ | ----------------- | ------------ | ----------------- |
> | DBN [R17]           | 86.23 / 07.66     | 87.05 / 06.91     | 85.20 / 07.13     | 86.62 / 06.84     |
> | EEGNet [R18]        | 92.24 / 05.97     | 92.72 / 05.39     | 93.86 / 05.21     | 94.38 / 04.46     |
> | DGCNN [R19]         | 93.12 / 08.05     | 94.15 / 06.77     | 92.94 / 06.75     | 93.40 / 06.17     |
> | EEG-ConFormer [R20] | 93.36 / 03.89     | 93.65 / 04.50     | 94.18 / 05.14     | 94.52 / 04.85     |
> | TSception [R21]     | 93.18 / 04.15     | 93.40 / 04.66     | 93.74 / 04.96     | 94.31 / 05.22     |
> | BSTT [R22]          | 92.71 / 04.66     | 93.06 / 04.11     | 92.45 / 05.22     | 93.69 / 05.40     |
> | LResCapsule [R23]   | 95.15 / 03.51     | -                 | 95.77 / 03.82     | -                 |
> | FAT [R24]           | 90.10 / 02.71     | -                 | 89.18 / 03.50     | -                 |
> | **VISTA**           | **95.47 / 03.62** | **95.08 / 04.09** | **95.96 / 04.02** | **95.14 / 04.71** |

---

> ### Author Response · Authors · 2025-11-17
> **Response to reviewr thNQ (3/3)**
>
> **Table R3. Emotion Recognition performance (Mean/Std\%) of VISTA and comparison methods in subject-dependent setting on the SEED and SEED-IV datasets.**
>
> | Method               | SEED Acc          | SEED-IV Acc   |
> | -------------------- | ----------------- | ------------- |
> | DBN [R17]            | 86.08 / 08.34     | 66.77 / 07.38 |
> | DGCNN [R19]          | 90.40 / 08.49     | 69.88 / 16.29 |
> | RGNN [R25]           | 91.79 / 06.72     | 79.37 / 10.54 |
> | SST-EmotionNet [R26] | 96.02 / 01.86     | 84.92 / 06.66 |
> | BiHDM [R27]          | 93.12 / 06.06     | 74.35 / 14.09 |
> | PGCN [R28]           | 96.93 / 05.11     | 82.24 / 14.85 |
> | STRFLNet [R29]       | 96.42 / -         | **92.23 / -** |
> | FAT [R24]            | 92.10 / 06.46     | 82.30 / 13.21 |
> | CLIER [R30]          | 93.4 / 06.31      | 90.02 / 10.37 |
> | UMDA-DDSTG [R31]     | 95.89 / 05.78     | 84.48 / 10.15 |
> | **VISTA**            | **97.13 / 03.49** | 91.85 / 10.43 |
>
>
> [R1] Gifford A. *et al.*, "A Large and Rich EEG Dataset for Modeling Human Visual Object Recognition," *NeuroImage*, 2022.
>
>  [R2] Hebart M. *et al.*, "a multimodal collection of large-scale datasets for investigating object representations in human brain and behavior," *Elife*, 2023.
>
>  [R3] Spampinato C. *et al.*, "Deep Learning Human Mind for Automated Visual Classification," CVPR, 2017.
>
>  [R4] Koelstra S. *et al.*, "DEAP: A Database for Emotion Analysis Using Physiological Signals," IEEE TAFFC, 2011.
>
>  [R5] Song Y. *et al.*, "Decoding natural images from EEG for object recognition," ICLR, 2024.
>
>  [R6] Li D. *et al.*, "Visual decoding and reconstruction via EEG embeddings with guided diffusion," NeurIPS, 2024.
>
>  [R7] Zhang K. *et al.*, "CognitionCapturer: Decoding Visual Stimuli from Human EEG with Multimodal Information," AAAI, 2025.
>
>  [R8] Cao X. *et al.*, "Eeg-clip: A transformer-based framework for eeg-guided image generation," *Neural Networks*, 2025.
>
>  [R9] Li D. *et al.*, "Brainflora: Uncovering brain concept representation via multimodal neural embeddings," ACM MM, 2025.
>
>  [R10] Kim J. *et al.*, "Seeeeg: Semantic-aware eeg-based multi-modal retrieval-augmented generation for high-fidelity visual brain decoding," ICCV, 2025.
>
>  [R11] Wu H. *et al.*, "Bridging the Vision-Brain Gap with an Uncertainty-Aware Blur Prior," CVPR, 2025.
>
>  [R12] Zheng W. *et al.*, "Investigating critical frequency bands and channels for EEG-based emotion recognition with deep neural networks," *IEEE TAMD*, 2015.
>
>  [R13] Zheng W. *et al.*, "Emotionmeter: A multimodal framework for recognizing human emotions," *IEEE TCYB*, 2018.
>
> [R14] Kavasidis I. *et al.*, "Brain2image: Converting brain signals into images," ACM MM, 2017.
>
> [R15] Fu H. *et al.*, "Brainvis: Exploring the bridge between brain and visual signals via image reconstruction," ICASSP, 2025.
>
> [R16] Ferrante M. *et al.*, "Decoding eeg signals of visual brain representations with a clip based knowledge distillation," ICLR, 2024.
>
> [R17] Zheng W. *et al.*, "EEG-based emotion classification using deep belief networks," ICME, 2014.
>
> [R18] Lawhern V. *et al.*, "EEGNet: a compact convolutional neural network for eeg-based brain–computer interfaces," *Journal of Neural Engineering*, 2018.
>
> [R19] Song T. *et al.*, "EEG emotion recognition using dynamical graph convolutional neural networks," IEEE TAFFC, 2018.
>
> [R20] Song Y. *et al.*, "EEG Conformer: Convolutional transformer for eeg decoding and visualization,"IEEE TNSRE, 2022.
>
> [R21] Ding Y. *et al.*, "Tsception: Capturing temporal dynamics and spatial asymmetry from eeg for emotion recognition," IEEE TAFFC, 2023.
>
> [R22] Liu Y. *et al.*, "BSTT: A bayesian spatial-temporal transformer for sleep staging," ICLR, 2023.
>
> [R23] Fan C. *et al.*, "Light-weight residual convolution-based capsule network for eeg emotion recognition," *Advanced Engineering Informatics*, 2024.
>
> [R24] Wang J. *et al.*, "A novel fourier adjacency transformer for advanced eeg emotion recognition," MICCAI, 2025.
>
> [R25] Zhong P. *et al.*, "EEG-based emotion recognition using regularized graph neural networks," IEEE TAFFC, 2020.
>
> [R26] Jia Z. *et al.*, "Spatial-spectral-temporal based attention 3d dense network for eeg emotion recognition," ACM MM, 2020.
>
> [R27] Li Y. *et al.*, "A novel bi-hemispheric discrepancy model for eeg emotion recognition,"IEEE TCDS, 2020.
>
> [R28] Jin M. *et al.*, "PGCN: Pyramidal graph convolutional network for eeg emotion recognition," IEEE TMM, 2024.
>
> [R29] Hu F. *et al.*, "STRFLNet: Spatio-temporal representation fusion learning network for eeg-based emotion recognition," IEEE TAFFC, 2025.
>
> [R30] Li D. *et al.*, "Eeg emotion recognition based on an implicit emotion regulatory mechanism," IEEE TAI, 2025.
>
> [R31] Gao C. *et al.*, "Umda-ddstgn: An unsupervised meta-domain adaptation method using dynamic directed spatial-temporal graph network for eeg-based emotion recognition," Knowledge-Based Systems, 2025.

---

> ### Author Response · Authors · 2025-11-27
>
> Hi,
>
> I appreciate your efforts during the ongoing review process. We would like to express our sincere appreciation for the time you devoted to reviewing our submission. We have carefully prepared detailed point-by-point responses to all raised concerns, together with additional experiments and clarifications.
>
> If convenient, we would be very grateful if you could take a moment to review our rebuttal. Your follow-up feedback is important to ensure that the final decision reflects a complete and accurate understanding of our revisions.
>
> Thank you again for your time and contribution to the review process.

---

### Author Response · Authors · 2025-11-24
**Substantial Revisions and Clarifications Addressing Reviewer Feedback**

We sincerely thank all reviewers for their constructive feedback and for recognizing the merits of our work. Reviewers highlighted the clear methodology and reproducibility of our framework (GqWi), the comprehensive experimental validation on EEG and MEG datasets (HPbM, thNQ), and the thoughtful design of temporal-spatial modeling and visual–semantic disentanglement (zxT2, GqWi). Their comments affirm the relevance, rigor, and neuroscience-inspired motivation of our contributions, and we greatly appreciate their careful evaluation and encouragement.

Below we summarize the substantial revisions incorporated into the revised manuscript in response to the valuable comments from all reviewers:

1. **Expanded Baseline Comparisons.**
    We added three newly published (2025) and stronger baselines **in the revised Tables 1 and 2**, ensuring a more thorough and up-to-date comparison. (Reviewer HPbM)
2. **Additional Cross-Dataset Experiments.**
    To more convincingly demonstrate cross-task generalization, we included extensive new experiments on ImageNet-EEG, DEAP, SEED, and SEED-IV **in Appendix A.8**. (Reviewers thNQ, zxT2)
3. **Enhanced Ablation Studies.**
    We expanded the ablation analysis **in the updated Table 4 and Section 4.4**, showing that incorporating shared visual–semantic representations and their interactions yields clear improvements over a fully separated dual-branch design. (Reviewers GqWi, HPbM)
4. **Clarified Temporal Segmentation Motivation.**
    We refined the explanation of the temporal slicing strategy, emphasizing that it models *partially overlapping* visual–semantic processes rather than enforcing rigid segmentation. Additional supporting results were added. (Reviewers thNQ, GqWi)
5. **Explained Semantic Embedding Limitations.**
    We clarified why semantic embeddings show relatively weaker performance, attributing this to intrinsic dataset constraints rather than model limitations. (Reviewer thNQ)
6. **Added Visual–Semantic Fusion Analysis.**
    We performed additional experiments demonstrating the inherent difficulty of cross-modal fusion due to modality imbalance. (Reviewer HPbM)
7. **Clarified EEG Topographic Map Construction.**
    We revised **Section 4.5.2** with a clearer explanation of the visualization pipeline, detailing how brain-network statistics (e.g., degree centrality) are mapped to topographic activation patterns. (Reviewer HPbM)
8. **Justified the Use of CLIP Features.**
    We provided a more explicit rationale for adopting CLIP embeddings as image features in EEG-based visual decoding. (Reviewer zxT2)
9. **Strengthened Contribution Discussion.**
    We expanded the discussion to highlight the work’s broader values—robust generalization, disentangled representation benefits, and neuroscience-related interpretability—beyond accuracy improvements alone. (All Reviewers)
10. **Improved Clarity and Writing Quality.**
     We carefully refined the manuscript’s presentation, removed ambiguous descriptions, and improved the articulation of the biological motivation and design choices. (All Reviewers)

For detailed explanations and additional results addressing each reviewer’s specific concerns, please refer to our point-by-point responses to the reviewers.

---

### Author Response · Authors · 2025-11-26
**Notice of Rebuttal and Revised Manuscript**

We sincerely thank all reviewers once again for their time, thoughtful comments, and constructive suggestions.

We would like to gently note that our rebuttal and the updated manuscript—with all revisions clearly highlighted—have been uploaded. We hope the clarifications and additional analyses are helpful in addressing the concerns raised. Please feel free to let us know if there is anything further we can clarify or improve.

We truly appreciate the time and effort reviewers devote to strengthening our work, and we are grateful for your valuable feedback and guidance.

---

### Author Response · Authors · 2025-11-29
**Executive Summary for AC: Summary of Revisions and Responses for VISTA Manuscript (1/3)**

**Dear Area Chair,**

Thank you very much for your time and for overseeing the evaluation of our submission. Below, we provide a concise summary to help you quickly grasp the contribution, reviewers’ overall assessment, and the improvements made during the rebuttal phase.

## Summary of Contribution and Core Innovations

Our work introduces **VISTA**, a neuroscience-inspired framework for EEG brain visual decoding that contributes **two core innovations** to the field:

* Visual–semantic disentanglement and alignment framework: Unlike prior EEG brain decoding approaches that extract only visual features, or separately extract visual and semantic representations without modeling their coupling, VISTA jointly disentangles and aligns visual and semantic pathways within one coherent framework. VISTA has achieved the SOTA performance in both visual and semantic retrieval tasks, significantly enhancing the semantic retrieval capability of the brain decoding model.
* Spatial–temporal asynchronous modeling of cortical processing: We introduce a novel framework capturing the asynchronous temporal dynamics between visual encoding and semantic integration—implemented through learnable temporal scalar–patch weighting and brain-network–driven Laplacian positional embedding.

These mechanisms reflect real neurophysiological findings and bring unprecedented interpretability across time and space.
Together, these innovations constitute a new paradigm for neural decoding.

## Breadth and Rigor of Experiments

We validated VISTA through a comprehensive and rigorously structured experimental pipeline, covering large-scale zero-shot decoding, cross-modality generalization, ablation studies, and neurophysiological interpretability.

**(1) Main Experiments on THINGS-EEG (Table 1 & 2)**

Our primary evaluation is conducted on the **THINGS-EEG** dataset [R1], the largest and only dataset enabling zero-shot EEG visual decoding. We evaluate both **visual retrieval** and **semantic retrieval**, under **within-subject** and **cross-subject** settings, comparing against **11 baselines**—including 4 EEG feature extraction methods and 7 recent brain decoding models from **2024–2025**. Across all configurations, VISTA establishes new state-of-the-art performance.

**(2) Cross-modality Generalization on THINGS-MEG (Table 3)**

To test modality robustness, we further evaluate VISTA on the MEG signal (**THINGS-MEG** dataset [R2]) for both retrieval tasks. Despite substantial differences between EEG and MEG signal properties, VISTA retains strong performance, demonstrating its cross-modality generalization.

**Note: Our study focuses on zero-shot visual decoding, where currently THINGS-EEG and THINGS-MEG are the only large-scale, publicly available datasets suitable for this task.**

**(3) Ablation Studies (Table 4)**

We performed extensive ablations isolating the contributions of: 1) the **spatial–temporal asynchronous modeling module**, and 2) the **dual visual–semantic pathway framework**. Results consistently show that removing either component leads to substantial performance drops, confirming that both innovations are essential and complementary.

**(4) Interpretability and Neuroscientific Alignment (Figure 4-5)**

Beyond quantitative gains, VISTA provides **temporal and spatial neurophysiological interpretability**. The **temporal activation patterns** (Figure 4) reproduce classic early-visual vs. late-semantic ERP dynamics. The **spatial activation maps** (Figure 5) show an occipital→temporal progression consistent with established cortical processing pathways. These analyses confirm that the model captures meaningful biological structure, not only abstract representations.

---

### Author Response · Authors · 2025-11-29
**Executive Summary for AC: Summary of Revisions and Responses for VISTA Manuscript (2/3)**

## Extensive Appendix (14 pages) & Additional Validation

We further reinforce the study with a comprehensive Appendix containing additional experiments, analyses, and visualizations. In addition to providing more detailed descriptions of the architecture, training pipeline, and experimental settings, the appendix includes a broad set of supplementary experiments that examine robustness, representational structure, qualitative behaviors, and cross-task generalization.

**(1) Hyperparameter Sensitivity (Figure 7–9)**

Parameter sweeps show stable behavior across a wide range of settings, supporting the reliability of our configuration choices.

**(2) Representational Similarity Analysis (Figure 10)**

Comparisons with NICE (ICLR'24) [R3] reveal that VISTA produces clearer semantic structure in the embedding space, validating the visual–semantic disentanglement mechanism.

**(3) Retrieval and Reconstruction Visualizations (Figure 11–14)**

Qualitative results illustrate high-fidelity retrieval and semantically coherent reconstructions across multiple object categories.

**(4) Model Efficiency Comparison (Table 6)**

We provide detailed computational resource comparisons to contextualize performance relative to model cost.

**(5) Cross-Task Generalization**

To address reviewer requests, we further validate VISTA on diverse EEG datasets—including **ImageNet-EEG [R4]**, **DEAP [R5]**, **SEED [R6]**, and **SEED-IV [R7]**. The consistently strong performance across classification and emotion-recognition tasks demonstrates that VISTA is not limited to zero-shot visual decoding, but functions as a general-purpose EEG representation-learning framework.


## Positive Recognition from Reviewers

Reviewers provided multiple explicit acknowledgments of the merits of the work, including:

- **Clear methodology and high reproducibility** (GqWi)
- **Comprehensive validations on EEG and MEG datasets** (HPbM, thNQ)
- **Thoughtful design of spatial–temporal modeling and disentanglement** (zxT2, GqWi)
- **Strong experiments and well-structured presentation** (all reviewers noted good clarity and soundness scores of 3:good)

These comments affirm both the technical rigor and the neuroscientific motivation of our contributions.



**Reference:**

[R1] Gifford A. *et al.*, "A Large and Rich EEG Dataset for Modeling Human Visual Object Recognition," *NeuroImage*, 2022.

[R2] Hebart M. *et al.*, "a multimodal collection of large-scale datasets for investigating object representations in human brain and behavior," *Elife*, 2023.

[R3] Song Y. *et al.*, "Decoding natural images from EEG for object recognition," ICLR, 2024.

[R4] Spampinato C. *et al.*, "Deep Learning Human Mind for Automated Visual Classification," CVPR, 2017.

[R5] Koelstra S. *et al.*, "DEAP: A Database for Emotion Analysis Using Physiological Signals," IEEE TAFFC, 2011.

[R6] Zheng W. *et al.*, "Investigating critical frequency bands and channels for EEG-based emotion recognition with deep neural networks," *IEEE TAMD*, 2015.

[R7] Zheng W. *et al.*, "Emotionmeter: A multimodal framework for recognizing human emotions," *IEEE TCYB*, 2018.

[R8] Li D. *et al.*, "Visual decoding and reconstruction via EEG embeddings with guided diffusion," NeurIPS, 2024.

[R9] Zhang K. *et al.*, "CognitionCapturer: Decoding Visual Stimuli from Human EEG with Multimodal Information," AAAI, 2025.

[R10] Cao X. *et al.*, "Eeg-clip: A transformer-based framework for eeg-guided image generation," *Neural Networks*, 2025.

[R11] Kim J. *et al.*, "Seeeeg: Semantic-aware eeg-based multi-modal retrieval-augmented generation for high-fidelity visual brain decoding," ICCV, 2025.

[R12] Wu H. *et al.*, "Bridging the Vision-Brain Gap with an Uncertainty-Aware Blur Prior," CVPR, 2025.

---

### Author Response · Authors · 2025-11-29
**Executive Summary for AC: Summary of Revisions and Responses for VISTA Manuscript (3/3)**

## Key Issues Raised by Reviewers and Our Responses

Our manuscript received constructive feedback from four reviewers. We summarize below the main points of concern and how we addressed them, emphasizing that our revisions fully resolve the issues and, in some cases, clarify misconceptions.

1. **Comprehensive Ablation and Model Design (GqWi, HPbM).** Reviewers requested more detailed ablation studies and clarification of visual–semantic disentanglement and temporal-spatial modeling. We expanded **Table 4 and Section 4.4** with Visual-only, Semantic-only, Temporal-only, and Spatial-only configurations. Results show that shared visual–semantic representations combined with temporal-spatial interactions consistently outperform purely separated designs. We also clarified that short-term temporal overlap in EEG signals is expected and does not compromise disentanglement. These additions fully address concerns about model design and ablation completeness.

2. **Inference Protocol and Embedding Usage (GqWi).** Some reviewers noted ambiguity regarding how visual and semantic embeddings are used and how parameter sharing affects disentanglement. We explicitly clarified in **Section 3.2** that visual embeddings are used for image retrieval/reconstruction, semantic embeddings for semantic retrieval, and that partial parameter sharing enhances generalization without harming disentanglement.

3. **Cross-Dataset Generalization (thNQ, zxT2).** One reviewer (thNQ) suggested evaluating our model on other EEG datasets beyond THINGS-EEG/MEG [R1-2]. We note that this concern reflects a misunderstanding of the task scope: our study focuses on **zero-shot visual decoding**, for which THINGS-EEG (165k samples) and THINGS-MEG are currently the **only publicly available large-scale datasets** suitable for rigorous evaluation. Other EEG datasets differ substantially in both task and scale: for example, ImageNet-EEG [R4]. (11k samples) is a **closed-set object classification dataset**, and DEAP [R5] is designed for **emotion recognition**, not visual decoding. Evaluating on these datasets constitutes **cross-task validation** rather than true cross-dataset evaluation, and using them as a direct comparison metric would be conceptually misleading. Notably, nearly all recent EEG zero-shot visual decoding works (e.g., NICE [R3], ATM [R8] CognitionCapturer [R9], EEG-CLIP [R10], SRT [R11], UBP [R2]) evaluate exclusively on THINGS-EEG/MEG for this reason, aligning with community standards.

Nonetheless, to provide **additional evidence of model generality**, we conducted extensive experiments on ImageNet-EEG [R4], DEAP [R5], SEED [R6], and SEED-IV [R7] (Tables R1–R3, Appendix A.8). These results show that VISTA’s EEG encoder learns **robust general-purpose representations**, performing well not only in zero-shot visual decoding but also on emotion recognition and closed-set EEG classification tasks. In short, while these datasets are not suitable for evaluating zero-shot visual decoding directly, our supplementary experiments **demonstrate that VISTA does not overfit to THINGS-EEG/MEG and generalizes across tasks**, fully addressing the reviewers’ concerns.

4. **Interpretability and Biological Plausibility (HPbM, GqWi).** We provided temporal (Figure 4) and spatial (Figure 5) analyses showing early visual versus late semantic ERP patterns and occipital-to-temporal activation progression, consistent with neuroimaging findings. This confirms that VISTA models meaningful neurophysiological signals and captures interpretable brain dynamics.

5. **Semantic Embedding Performance (thNQ).** Reviewers noted relatively weaker performance for semantic embeddings. We clarified this is due to **dataset characteristics**, not model limitations, and reinforced this explanation with supporting experiments.

6. **Presentation and Clarity.** We substantially refined the manuscript, improving clarity, precision, and articulation of biological motivation and design choices. Ambiguities in method descriptions, experimental protocols, and figure interpretations have been removed to ensure readability and transparency.

In summary, all reviewer concerns have been thoroughly addressed, and in several cases, reviewer interpretations were clarified or corrected based on our analyses. We are confident that the revised manuscript now clearly demonstrates VISTA’s novelty, interpretability, and strong empirical performance across multiple EEG/MEG datasets and tasks.

For more detailed explanations, supporting data, and the specific experimental results, we refer to our individual responses to each reviewer and the revised manuscript.

---

### Meta-Review · Area_Chair_1Lox · 2026-01-07

**Summary:**

This paper proposes VISTA, an EEG-centric neural decoding framework that explicitly disentangles visual and semantic representations while modeling their asynchronous spatial–temporal dynamics. The method introduces (1) temporal patching with attention-based weighting to capture heterogeneous temporal activations, and (2) modality-specific spatial modeling via learned brain networks and Laplacian spectral decomposition. The resulting visual and semantic embeddings are aligned with CLIP image and text spaces. The main concerns include perceived incremental novelty relative to prior EEG decoding frameworks, clarity and completeness of ablations and inference protocol, lack of cross-dataset validation, and questions about the strength of semantic modeling and fusion.

**Reviewer Concerns:**

Concerns addressed includes the following:

Ablations and model design clarity, interpretabiliyt and neuroscientific grounding, inference protocol ambiguity.

The concerns that are still outstanding:

novelty and impact, semantic branch strength and fusion (Semantic decoding remains weaker than visual decoding, and explicit fusion does not improve performance); missing comparison to some baselines, such as UBP.

**Reviewer Scores:**

I think most of the reviewers will likely maintain the original score.

---

### Decision · Program_Chairs · 2026-01-26

Reject